# PROTEIN STRUCTURE GENERATION VIA FOLDING DIFFUSION

## ABSTRACT

The ability to computationally generate novel yet physically foldable protein structures could lead to new biological discoveries and new treatments targeting yet incurable diseases. Despite recent advances in protein structure prediction, directly generating diverse, novel protein structures from neural networks remains difficult. In this work, we present a new diffusion-based generative model that designs protein backbone structures via a procedure that mirrors the native folding process. We describe protein backbone structure as a series of consecutive angles capturing the relative orientation of the constituent amino acid residues, and generate new structures by denoising from a random, unfolded state towards a stable folded structure. Not only does this mirror how proteins biologically twist into energetically favorable conformations, the inherent shift and rotational invariance of this representation crucially alleviates the need for complex equivariant networks. We train a denoising diffusion probabilistic model with a simple transformer backbone and demonstrate that our resulting model unconditionally generates highly realistic protein structures with complexity and structural patterns akin to those of naturally-occurring proteins. As a useful resource, we release the first open-source codebase and trained models for protein structure diffusion.

## 1 INTRODUCTION

Proteins are critical for life, playing a role in almost every biological process, from relaying signals across neurons (Zhou et al., 2017) to recognizing microscopic invaders and subsequently activating the immune response (Mariuzza et al., 1987), from producing energy for cells (Bonora et al., 2012) to transporting molecules along cellular highways (Dominguez & Holmes, 2011). Misbehaving proteins, on the other hand, cause some of the most challenging ailments in human healthcare, including Alzheimer's disease, Parkinson's disease, Huntington's disease, and cystic fibrosis (Chaudhuri & Paul, 2006). Due to their ability to perform complex functions with high specificity, proteins have been extensively studied as a therapeutic medium (Leader et al., 2008; Kamionka, 2011; Dimitrov, 2012) and constitute a rapidly growing segment of approved therapies (H Tobin et al., 2014). Thus, the ability to computationally generate novel yet physically foldable protein structures could open the door to discovering novel ways to harness cellular pathways and eventually lead to new treatments targeting yet incurable diseases.

Many works have tackled the problem of computationally generating new protein structures, but have generally run into challenges with creating diverse yet realistic folds. Traditional approaches typically apply heuristics to assemble fragments of experimentally profiled proteins into structures (Schenkelberg & Bystroff, 2016; Holm & Sander, 1991). This approach is limited by the boundaries of expert knowledge and available data. More recently, deep generative models have been proposed. However, due to the incredibly complex structure of proteins, these commonly do not directly generate protein structures, but rather constraints (such as pairwise distance between residues) that are heavily post-processed to obtain structures (Anand et al., 2019; Lee & Kim, 2022). Not only does this add complexity to the design pipeline, but noise in these predicted constraints can also be compounded during post-processing, resulting in unrealistic structures – that is, if the constraints are at all satisfiable to begin with. Other generative models rely on complex equivariant network architectures or loss functions to learn to generate a 3D point cloud that describes a protein structure (Anand & Achim, 2022; Trippe et al., 2022; Luo et al., 2022; Eguchi et al., 2022). Such equivariant architectures can ensure that the probability density from which the protein structures are sampled is

invariant under translation and rotation. However, translation- and rotation-equivariant architectures are often also symmetric under reflection, leading to violations of fundamental structural properties of proteins like chirality (Trippe et al., 2022). Intuitively, this point cloud formulation is also quite detached from how proteins biologically fold – by twisting to adopt energetically favorable configurations (Šali et al., 1994; Englander et al., 2007).

Inspired by the *in vivo* protein folding process, we introduce a generative model that acts on the *inter-residue angles* in protein backbones instead of on Cartesian atom coordinates (Figure 1). This treats each residue as an independent reference frame, thus shifting the equivariance requirements from the neural network to the coordinate system itself. A similar angular representation has been used in some protein structure prediction works (Gao et al., 2017; AlQuraishi, 2019; Chowdhury et al., 2022). For generation, we use a denoising diffusion probabilistic model (diffusion model, for brevity) (Ho et al., 2020; Sohl-Dickstein et al., 2015) with a vanilla transformer parameterization without any equivariance constraints. Diffusion models train a neural network to start from noise and iteratively "denoise" it to generate data samples. Such models have been highly successful in a wide range of data modalities from images (Saharia et al., 2022; Rombach et al., 2022) to audio (Rouard & Hadjeres, 2021; Kong et al., 2021), and are easier to train with better modal coverage than methods like generative adversarial networks (GANs) (Dhariwal & Nichol, 2021; Nichol & Dhariwal, 2021). We present a suite of validations to quantitatively demonstrate that unconditional sampling from our model directly generates realistic protein backbones – from recapitulating the natural distribution of protein inter-residue angles, to producing overall structures with appropriate arrangements of multiple structural building block motifs. We show that our generated backbones are diverse and designable, and are thus biologically plausible protein structures. Our work demonstrates the power of biologically-inspired problem formulations and represents an important step towards accelerating the development of new proteins and protein-based therapies.

## 2 Related work

### 2.1 Generating new protein structures

Many generative deep learning architectures have been applied to the task of generating novel protein structures. Anand et al. (2019) train a GAN to sample pairwise distance matrices that describe protein backbone arrangements. However, these pairwise distance matrices must be corrected, refined, and converted into realizable backbones via two independent post-processing steps, the Alternating Direction Method of Multipliers (ADMM) and Rosetta. Crucially, inconsistencies in these predicted constraints can render them unsatisfiable or lead to significant errors when reconstructing the final protein structure. Sabban & Markovsky (2020) use a long short-term memory (LSTM) GAN to generate $(\phi, \psi)$ dihedral angles. However, their network only generates $\alpha$ helices and relies on downstream post-processing to filter, refine, and fold structures, partly due to the fact that these two dihedrals do not sufficiently specify backbone structure. Eguchi et al. (2022) propose a variational auto-encoder with equivariant losses to generate protein backbones in 3D space. However, their work only targets immunoglobulin proteins and also requires refinement through Rosetta. Non-deep learning methods have also been explored: Schenkelberg & Bystroff (2016) apply heuristics to ensembles of similar sequences to perturb known protein structures, while Holm & Sander (1991) use a database search to find and assemble existing protein fragments that might fit a new scaffold structure. These approaches' reliance on known proteins and hand-engineered heuristics limit them to relatively small deviations from naturally-occurring proteins.

#### 2.1.1 Diffusion models for protein structure generation

Several recent works have proposed extending diffusion models towards generating protein structures. These predominantly perform diffusion on the 3D Cartesian coordinates of the residues themselves. For example, Trippe et al. (2022) use an E(3)-equivariant graph neural network to model the coordinates of protein residues. Anand & Achim (2022) adopt a hybrid approach where they train an equivariant transformer with invariant point attention (Jumper et al., 2021); this model generates the 3D coordinates of $C_\alpha$ atoms, the amino acid sequence, and the angles defining the orientation of side chains. Another recent work by Luo et al. (2022) performs diffusion for generating antibody fragments' structure and sequence by modeling 3D coordinates using an equivariant neural network. Note that these prior works all use some form of equivariance to translation, rotation,

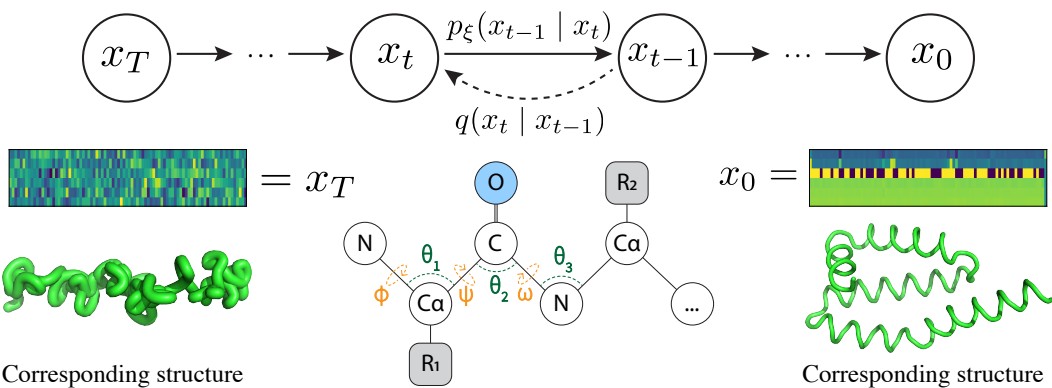

Figure 1: We perform diffusion on six angles as illustrated in the schematic in the bottom center (also defined in Table 1). Three of these are dihedral torsion angles (orange), and three are bond angles (green). We start with an experimentally observed backbone described by angles $x_0$ and iteratively add Gaussian noise via the forward noising process $q$ until the angles are indistinguishable from a wrapped Gaussian at $x_T$. We use these examples to learn the "reverse" denoising process $p_\xi$.

and/or reflection due to their formulation of diffusion on Cartesian coordinates. Another method, ProteinSGM (Lee & Kim, 2022), implements a score-based diffusion model (Song et al., 2020) that generates image-like square matrices describing pairwise angles and distances between all residues in an amino acid chain. However, this set of values is highly over-constrained, and must be used as a set of *input constraints* for Rosetta's folding algorithm (Yang et al., 2020), which in turn produces the final folded output. This is a similar approach to Anand et al. (2019), and is likewise subject to the aforementioned concerns regarding complexity, satisfiability, and cleanliness of predicted constraints. Our work instead uses a minimal set of angles required to specify a protein backbone, and thus directly generates structures *without* relying on additional methods for refinement. Unfortunately, none of these prior works have publicly-available code, model weights, or generated examples at the time of this writing. Thus, our ability to perform direct comparisons is limited.

## 2.2 DIFFUSION MODELS FOR SMALL MOLECULES

A related line of work focuses on creating and modeling small molecules, typically in the context of drug design, using similar generative approaches. These small molecules average 44 atoms in size (Jing et al., 2022). Compared to proteins, which average several hundred residues and thousands of atoms (Tiessen et al., 2012), the relatively small size of small molecules makes them easier to model. The E(3) Equivariant Diffusion Model (Hoogeboom et al., 2022) uses an equivariant transformer to design small molecules by diffusing their coordinates in Euclidean space. Other works have explored torsional diffusion, i.e., modelling the angles that specify a small molecule, to sample from the space of energetically favorable molecular conformations (Jing et al., 2022). This work still requires an $SE(3)$-equivariant model as the input to their model is a 3D point cloud. In contrast, our problem formulation allows us to work entirely in terms of relative angles.

## 3 METHOD

### 3.1 SIMPLIFIED FRAMING OF PROTEIN BACKBONES USING INTERNAL ANGLES

Proteins are variable-length chains of amino acid residues. There are 20 canonical amino acids, all of which share the same three-atom $N - C_\alpha - C$ backbone, but have varying side chains attached to the $C_\alpha$ atom (typically denoted $R_i$, see illustration in Figure 1). These residues assemble to form polymer chains typically hundreds of residues long (Tiessen et al., 2012). These chains of amino acids fold into 3D structures, taking on a shape that largely determines the protein's functions. These folded structures can be described on four levels: primary structure, which simply captures the linear sequence of amino acids; secondary structure, which describes the *local* arrangement of amino acids and includes structural motifs like $\alpha$-helices and $\beta$-sheets; tertiary structure, which describes the full

Table 1: Internal angles used to specify protein backbone structure. Some of these involve multiple residues, indicated via $i$ index subscripts. These are illustrated in Figure 1.

| Angle | Description |
|---|---|
| $\psi$ | Dihedral torsion about $N_i - C\alpha_i - C_i - N_{i+1}$ |
| $\omega$ | Dihedral torsion about $C\alpha_i - C_i - N_{i+1} - C\alpha_{i+1}$ |
| $\phi$ | Dihedral torsion about $C_i - N_{i+1} - C\alpha_{i+1} - C_{i+1}$ |
| $\theta_1$ | Bond angle about $N_i - C\alpha_i - C_i$ |
| $\theta_2$ | Bond angle about $C\alpha_i - C_i - N_{i+1}$ |
| $\theta_3$ | Bond angle about $C_i - N_{i+1} - C\alpha_{i+1}$ |

spatial arrangement of all residues; and quaternary structure, which describes how multiple different amino acid chains come together to form larger complexes (Sun et al., 2004).

We propose a simplified framing of protein backbones that follows the biological intuition of protein folding while removing the need for complex equivariant networks. Rather than viewing a protein backbone of length $N$ amino acids as a cloud of 3D coordinates (i.e., $x \in \mathbb{R}^{N \times 3}$ if modeling only $C_\alpha$ atoms, or $x \in \mathbb{R}^{3N \times 3}$ for a full backbone) as prior works have done, we view it as a sequence of six internal, consecutive angles $x \in [-\pi, \pi)^{(N-1) \times 6}$. That is, each vector of six angles describes the relative position of all backbone atoms in the *next* residue given the position of the *current* residue. These six angles are defined precisely in Table 1 and illustrated in Figure 1. These internal angles can be easily computed using trigonometry, and converted back to 3D Cartesian coordinates by iteratively adding atoms to the protein backbone as described in Parsons et al. (2005), fixing bond distances to average lengths (Appendix A.1, Figure S1). Despite building the structure from a fixed point, this formulation does not appear to overly accumulate errors (Appendix A.2, Figures S2, S3).

This internal angle formulation has several key advantages. Most importantly, since each residue forms its own independent reference frame, there is no need to use an equivariant neural network. No matter how the protein is rotated or shifted, the angles specifying the *next* residue given the *current* residue never changes. This allows us to use a simple transformer as the backbone architecture; in fact, we demonstrate that our model fails when substituting our shift- and rotation-invariant internal angle representation with Cartesian coordinates, keeping all other design choices identical (Appendix C.1, Figure S4). This internal angle formulation also closely mimics how proteins actually fold by twisting into more energetically stable conformations.

## 3.2 DENOISING DIFFUSION PROBABILISTIC MODELS

Denoising diffusion probabilistic models (or diffusion models, for short) leverage a Markov process $q(x_t \mid x_{t-1})$ to corrupt a data sample $x_0$ over $T$ discrete timesteps until it is indistinguishable from noise at $x_T$. A diffusion model $p_\xi(x_{t-1} \mid x_t)$ parameterized by $\xi$ is trained to reverse this forward noising process, "denoising" pure noise towards samples that appear drawn from the native data distribution (Sohl-Dickstein et al., 2015). Diffusion models were first shown to achieve good generative performance by Ho et al. (2020); we adapt this framework for generating protein backbones, introducing necessary modifications to work with periodic angular values.

We modify the standard Markov forward noising process that adds noise at each discrete timestep $t$ to sample from a wrapped normal instead of a standard normal (Jing et al., 2022):

$$q(x_t \mid x_{t-1}) = \mathcal{N}_{\text{wrapped}}(x_t; \sqrt{1 - \beta_t} x_{t-1}, \beta_t I) \propto \sum_{k=-\infty}^{\infty} \exp \left( \frac{-\|x_t - \sqrt{1 - \beta_t} x_{t-1} + 2\pi k\|^2}{2\beta_t^2} \right)$$

where $\beta_t \in (0, 1)_{t=1}^T$ are set by a variance schedule. We use the cosine variance schedule (Nichol & Dhariwal, 2021) with $T = 1000$ timesteps:

$$\beta_t = \text{clip}\left(1 - \frac{\bar{\alpha}_t}{\bar{\alpha}_{t-1}}, 0.999\right) \quad \bar{\alpha}_t = \frac{f(t)}{f(0)} \quad f(t) = \cos\left(\frac{t/T + s}{1 + s} \cdot \frac{\pi}{2}\right)$$

where $s = 8 \times 10^{-3}$ is a small constant for numerical stability. We train our model for $p_\xi(x_{t-1}|x_t)$ with the simplified loss proposed by Ho et al. (2020), using a neural network $\text{nn}_\xi(x_t, t)$ that predicts the noise $\epsilon \sim \mathcal{N}(0, I)$ present at a given timestep (rather than the denoised mean values themselves). To handle the periodic nature of angular values, we introduce a function to "wrap" values within the range $[-\pi, \pi)$: $w(x) = ((x + \pi) \mod 2\pi) - \pi$. We use $w$ to wrap a smooth L1 loss (Girshick, 2015) $L_w$, which behaves like L1 loss when error is high, and like an L2 loss when error is low; we set the transition between these two regimes at $\beta_L = 0.1\pi$. While this loss is not as well-motivated as torsional losses proposed by Jing et al. (2022), we find that it achieves strong empirical results.

$$d_w = w\left(\epsilon - \text{nn}_\xi\left(w\left(\sqrt{\bar{\alpha}_t}x_0 + \sqrt{1 - \bar{\alpha}_t}\epsilon\right), t\right)\right)$$

$$L_w = \begin{cases} 0.5\frac{d_w^2}{\beta_L} & \text{if } |d_w| < \beta_L \\ |d_w| - 0.5\beta_L & \text{otherwise} \end{cases}$$

During training, timesteps are sampled uniformly $t \sim U(0, T)$. We normalize all angles in the training set to be zero mean by subtracting their element-wise angular mean $\mu$; validation and test sets are shifted by this same offset.

Figure 1 illustrates this overall training process, including our previously described internal angle framing. The internal angles describing the folded chain $x_0$ are corrupted until they become indistinguishable from random angles, which results in a disordered mass of residues at $x_T$; we sample points along this diffusion process to train our model $\text{nn}_\xi$. Once trained, the reverse process of sampling from $p_\xi$ also requires modifications to account for the periodic nature of angles, as described in Algorithm 1. The variance of this reverse process is given by $\sigma_t = \sqrt{\frac{1 - \bar{\alpha}_{t-1}}{1 - \bar{\alpha}_t} \cdot \beta_t}$.

---

**Algorithm 1** Sampling from $p_\xi$ with FoldingDiff

---

1: $x_T \sim w(\mathcal{N}(0, I))$               ▷ Sample from a wrapped Gaussian
2: **for** $t = T, \dots, 1$ **do**
3:      $z = \mathcal{N}(0, I)$ if $t > 1$ else $z = 0$
4:      $x_{t-1} = w\left(\frac{1}{\sqrt{\alpha_t}}\left(x_t - \frac{1 - \alpha_t}{\sqrt{1 - \bar{\alpha}_t}}\text{nn}_\xi(x_t, t)\right) + \sigma_t z\right)$     ▷ Wrap sampled values about $[-\pi, \pi)$
5: **end for**
6: **return** $w(x_0 + \mu)$               ▷ Un-shift generated values by original mean shift

---

This sampling process can be intuitively described as refining internal angles from an unfolded state towards a folded state. As this is akin to how proteins fold *in vivo*, we name our method FoldingDiff.

### 3.3 MODELING AND DATASET

For our reverse (denoising) model $p_\xi(x_t, t)$, we adopt a vanilla bidirectional transformer architecture (Vaswani et al., 2017) with relative positional embeddings (Shaw et al., 2018). Our six-dimensional input is linearly upscaled to the model's embedding dimension ($d = 384$). To incorporate the timestep $t$, we generate random Fourier feature embeddings (Tancik et al., 2020) as done in Song et al. (2020) and add these embeddings to each upscaled input. To convert the transformer's final per-position representations to our six outputs, we apply a regression head consisting of a densely connected layer, followed by GELU activation (Hendrycks & Gimpel, 2016), layer normalization, and finally a fully connected layer outputting our six values. We train this network with the AdamW optimizer (Loshchilov & Hutter, 2019) over 10,000 epochs, with a learning rate that linearly scales from 0 to $5 \times 10^{-5}$ over 1,000 epochs, and back to 0 over the final 9,000 epochs. Validation loss appears to plateau after $\approx 1,400$ epochs; additional training does not improve validation loss, but appears to lead to a poorer diversity of generated structures. We thus take a model checkpoint at 1,488 epochs for all subsequent analyses.

We train our model on the CATH dataset, which provides a "de-duplicated" set of protein structural folds spanning a wide range of functions where no two chains share more than 40% sequence identity over 60% overlap (Sillitoe et al., 2015). We exclude any chains with fewer than 40 residues. Chains longer than 128 residues are randomly cropped to a 128-residue window at each epoch. A random

80/10/10 training/validation/test split yields 24,316 training backbones, 3,039 validation backbones, and 3,040 test backbones.

# 4 EXPERIMENTS

## 4.1 GENERATING PROTEIN INTERNAL ANGLES

After training our model, we check that it is able to recapitulate the correct marginal distributions of dihedral and bond angles in proteins. We unconditionally generate 10 backbone chains each for every length $l \in [50, 128]$, generating a total of 780 backbones as was done in Trippe et al. (2022). We plot the distributions of all six angles, aggregated across these 780 structures, and compare each distribution to that of experimental test set structures less than 128 residues in length (Figures 2, S9). We observe that, across all angles, the generated distribution almost exactly recapitulates the test distribution. This is true both for angles that are nearly Gaussian with low variance $(\omega, \theta_1, \theta_2, \theta_3)$ as well as for angles with highly complex, high-variance distributions $(\phi, \psi)$. Angles that wrap about the $-\pi/\pi$ boundary $(\omega)$ are correctly handled as well. Compared to similar plots generated from other protein diffusion methods (e.g., Figure 1 in Anand & Achim (2022), reproduced with permission in Figure S10), we qualitatively observe that our method produces a much tighter distribution that more closely matches the natural distribution of bond angles.

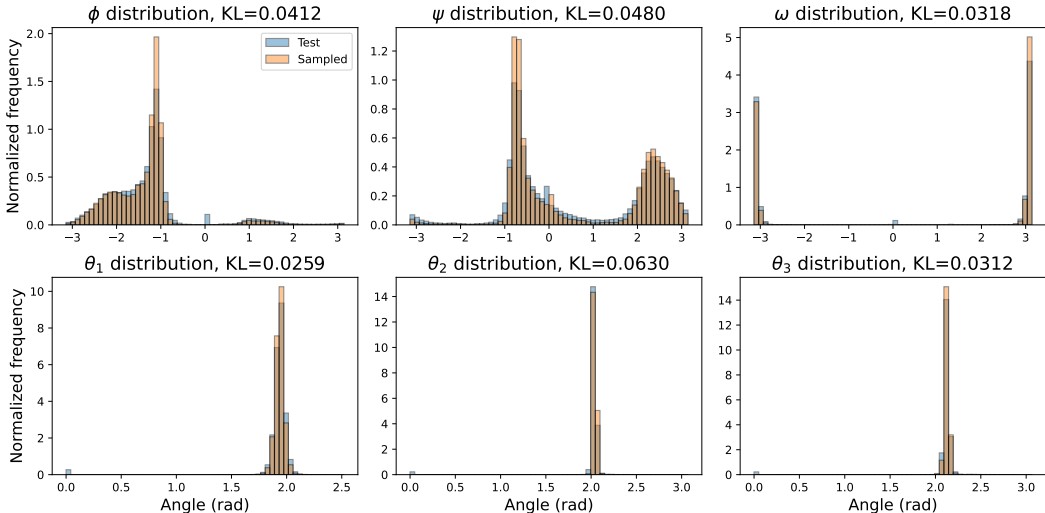

Figure 2: Comparison of the distributions of angular values in held-out test set and in generated samples. Top row shows dihedral angles (torsional angles involving 4 atoms), and bottom row shows bond angles (involving 3 atoms). KL divergence is calculated between $D_{KL}(\text{sampled}||\text{test})$. Figure S9 shows the cumulative distribution function (CDF) corresponding to these histograms.

However, looking at individual distributions of angles alone does not capture the fact that these angles are not independently distributed, but rather exhibit significant correlations. A Ramachandran plot, which shows the frequency of co-occurrence between the dihedrals $(\phi, \psi)$, is commonly used to illustrate these correlations between angles (Ramachandran & Sasisekharan, 1968). Figure 3 shows the Ramachandran plot for (experimental) test set chains with fewer than 128 residues, as well as that for our 780 generated structures. The Ramachandran plot for natural structures (Figure 3a) contains three major concentrated regions corresponding to right-handed $\alpha$ helices, left-handed $\alpha$ helices, and $\beta$ sheets. All three of these regions are recapitulated in our generated structures (Figure 3b). In other words, FoldingDiff is able to generate all three major secondary structure elements in protein backbones. Furthermore, we see that our model correctly learns that right-handed $\alpha$ helices are much more common than left-handed $\alpha$ helices (Cintas, 2002). Prior works that use equivariant networks, such as Trippe et al. (2022), cannot differentiate between these two types of helices due to network equivariance to reflection. This concretely demonstrates that our internal angle formulation leads to improved handling of chirality (i.e., the asymmetric nature of proteins) in generated backbones.

(a) Ramachandran plot, test set

(b) Ramachandran plot, generated backbones

Figure 3: Ramachandran plots comparing the $(\phi, \psi)$ dihedral angles for test set (3a) and generated protein backbones (3b). Each major region of this plot indicates a different secondary structure element, as indicated in panel 3a. All three main structural elements are recapitulated in our generated backbones, along with some less common angle combinations. Lines are artifacts of null values, and appear shifted due to zero centering/uncentering.

## 4.2 ANALYZING GENERATED STRUCTURES

We have shown that our model generates realistic distributions of angles and that our generated joint distributions capture secondary structure elements. We now demonstrate that the overall structures specified by these angles are biologically reasonable. Recall that natural protein structures contain multiple secondary structure elements. We use P-SEA (Labesse et al., 1997) to count the number of secondary structure elements in each test-set backbone of fewer than 128 residues, and generate a 2D histogram describing the frequency of $\alpha/\beta$ co-occurrence counts in Figure 4a. Figure 4b repeats this analysis for our generated structures, which frequently contain multiple secondary structure elements just as naturally-occurring proteins do. FoldingDiff thus appears to generate rich structural information, with consistent performance across multiple generation replicates (Figure S11). This is a nontrivial task – an autoregressive transformer, for example, collapses to a failure mode of endlessly generating $\alpha$ helices (Appendix C.2, Figures S5, S6).

(a) Secondary structure co-occurrence, test

(b) Secondary structure co-occurrence, generated

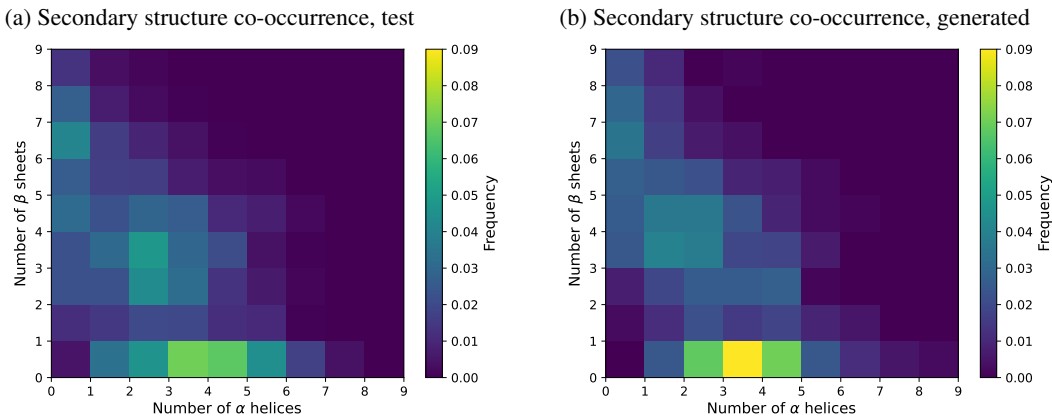

Figure 4: 2D histograms describing co-occurrence of secondary structures in test backbones (4a) and generated backbones (4b). Axes indicate the number of secondary structure present in a chain; color indicates the frequency of a specific combination of secondary structure elements. Our generated structures mirror real structures with multiple $\alpha$ helices, multiple $\beta$ sheets, and a mixture of both. See Figure S11 for additional generation replicates.

Beyond demonstrating that FoldingDiff's generated backbones contain reasonable structural motifs, it is also important to show that they are designable – meaning that we can find a sequence of amino acids that can fold into our designed backbone structure. After all, a novel protein structure is not useful if we cannot physically realize it. Previous works evaluate this *in silico* by predicting possible amino acids that fold into a generated backbone and checking whether the predicted structure for these sequences matches the original backbone (Trippe et al. 2022, Lee & Kim 2022, Appendix B). Following this general procedure, for a generated structure $s$, we use the ProteinMPNN inverse folding model (Dauparas et al., 2022) to generate 8 different amino acid sequences, as it yields improved performance compared to ESM-IF1 (Hsu et al. 2022, Appendix C.3, Tables S1, S2). We then use OmegaFold (Wu et al., 2022) to predict the 3D structures $\hat{s}_1, \ldots, \hat{s}_8$ corresponding to each of these sequences. We use TMalign (Zhang & Skolnick, 2005), which evaluates structural similarity between backbones, to score each of these 8 structures against the original structure $s$. The maximum score $\max_{i \in [1,8]} \mathrm{TMalign}(s, \hat{s}_i)$ is the self-consistency TM (scTM) score. A scTM score of $\geq 0.5$ is considered to be in the same fold, and thus is "designable."

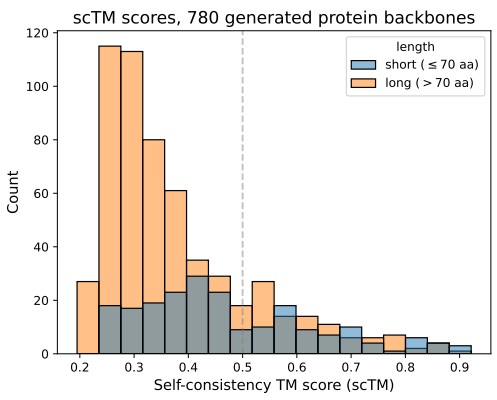

(a) Backbone designability by length

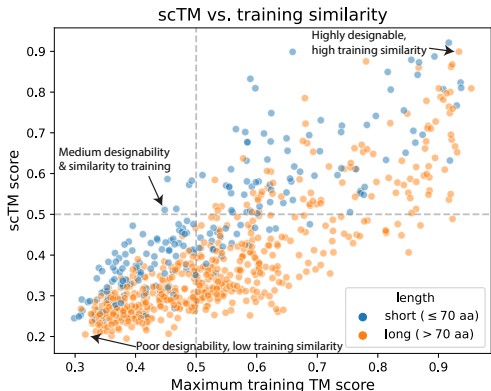

(b) Designability compared to training set similarity

Figure 5: Of our 780 generated backbones, ranging in length from 50-128 residues, 177 are designable ($\mathrm{scTM} \geq 0.5$) using ProteinMPNN and OmegaFold. Shorter structures of 70 amino acids or fewer tend to have higher scTM scores than longer structures (5a). Generated backbones that are more similar to training examples (greater maximum training TM score) tend to have better designability (5b). The three structures indicated by arrows are illustrated in Figure S12.

With this procedure, we find that 177 of our 780 structures, or 22.7%, are designable with an scTM score $\geq 0.5$ (Figure 5a) without any refinement or relaxation. This designability is highly consistent across different generation runs (Table S1), and is also consistent when substituting AlphaFold2 without MSAs (Jumper et al., 2021) in place of OmegaFold (163/780 designable with AlphaFold2). Trippe et al. (2022) use an identical scTM pipeline using ProteinMPNN and AlphaFold2, and report a significantly lower proportion of designable structures (92/780 designable, $p \ll 10^{-5}$, Chi-square test). Compared to this prior work, FoldingDiff improves upon designability of both short sequences (up to 70 residues, 76/210 designable compared to 36/210, $p = 1 \times 10^{-5}$, Chi-square test) and long sequences (beyond 70 residues, 87/570 designable compared to 56/570, $p = 5.6 \times 10^{-3}$, Chi-square test). While ProteinSGM (ESM-IF1 for inverse folding, AlphaFold2 for fold prediction) reports an even higher designability proportion of 50.5%, this value is not directly comparable, as ProteinSGM generates *constraints* that are subsequently folded using Rosetta; therefore, their designability does not directly reflect their generative process. The authors themselves note that Rosetta "post-processing" significantly improves the viability of their structures. To further contextualize our scTM scores, we evaluate a naive method that samples from the empirical distribution of dihedral angles. This baseline produces no designable structures whatsoever (Appendix C.3, Figures S7, S8). Conversely, we evaluate experimental structures to establish an upper bound for designability; 87% of natural structures have an scTM $\geq 0.5$ (Appendix C.3, Figure S7).

We additionally evaluate the similarity of each generated backbone to any training backbone by taking the maximum TM score across the entire training set. The maximum training TM-score is significantly correlated with scTM score (Spearman's $r = 0.78$, $p = 7.9 \times 10^{-165}$, Figure 5b), indi-

cating that structures more similar to the training set tend to be more designable. However, this does not suggest that we are merely memorizing the training set; doing so would result in a distribution of training TM scores near 1.0, which is not what we observe. ProteinSGM reports a distribution of training set TMscores much closer to 1.0; this suggests a greater degree of memorization and may indicate that their high designability ratio is partially driven by memorization.

Selected examples of our generated backbones and corresponding OmegaFold predictions of various lengths are visualized using PyMOL (Schrödinger, LLC, 2015) in Figure 6. Interestingly, we find that of our 177 designable backbones, only 16 contain $\beta$ sheets as annotated by P-SEA. Conversely, of our 603 backbones with scTM $< 0.5$, 533 contain $\beta$ sheets. This suggests that generated structures with $\beta$ sheets may be less designable ($p \ll 1.0 \times 10^{-5}$, Chi-square test). We additionally cluster our designable backbones and observe a large diversity of structures (Figure S14) comparable to that of natural structures (Figure S15). This suggests that our model is not simply generating small variants on a handful of core structures, which prior works appear to do (Figure S16).

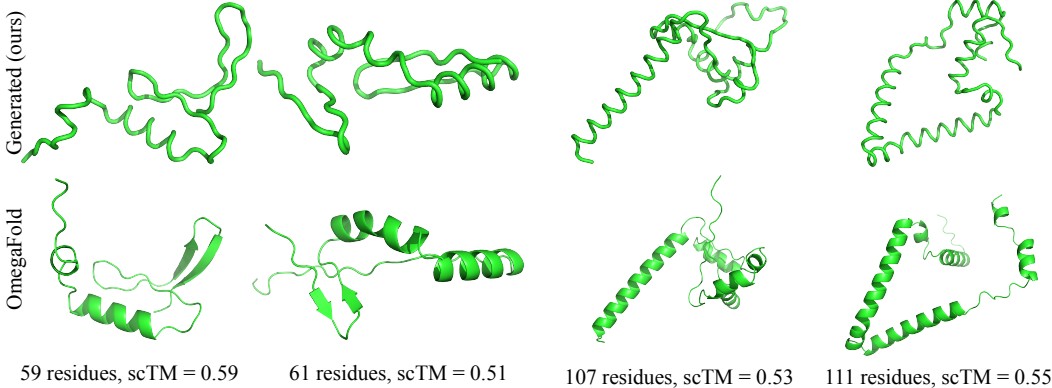

59 residues, scTM = 0.59        61 residues, scTM = 0.51        107 residues, scTM = 0.53        111 residues, scTM = 0.55

Figure 6: Selected generated protein backbones of varying length that are approximately designable (scTM $\approx 0.5$). Top row shows our directly generated backbones; bottom row shows OmegaFold predicted structure for residues inferred by ProteinMPNN to produce our generated backbone. Structures contain both $\alpha$ helices (coils, columns 1-4) and $\beta$ sheets (ribbons, columns 1 and 2), and each appears meaningfully different from its most similar training example (Figure S13).

## 5    CONCLUSION

In this work, we present a novel parameterization of protein backbone structures that allows for simplified generative modeling. By considering each residue to be its own reference frame, we describe a protein using the resulting relative internal angle representation. We show that a vanilla transformer can then be used to build a diffusion model that generates high-quality, biologically plausible, diverse protein structures. These generated backbones respect protein chirality and exhibit high designability.

While we demonstrate promising results with our model, there are several limitations to our work. Though formulating a protein as a series of angles enables use of simpler models without equivariance mechanisms, this framing allows for errors early in the chain to significantly alter the overall generated structure – a sort of "lever arm effect." Additionally, some generated structures exhibit collisions where the generated structure crosses through itself. Future work could explore methods to avoid these pitfalls using geometrically-informed architectures such as those used in Wu et al. (2022). Our generated structures are still of relatively short lengths compared to natural proteins which typically have several hundred residues; future work could extend towards longer structures, potentially incorporating additional losses or inputs that help "checkpoint" the structure and reduce accumulation of error. We also do not handle multi-chain complexes or ligand interactions, and are only able to generate static structures that do not capture the dynamic nature of proteins. Future work could incorporate amino acid sequence generation in parallel with structure generation, along with guided generation using functional or domain annotations. In summary, our work provides an important step in using biologically-inspired problem formulations for generative protein design.

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

# A   INTERNAL ANGLE FORMULATION OF PROTEIN BACKBONES

## A.1   CHOICE OF ANGLES FOR REPRESENTATION

A protein backbone structure can be fully specified by a total of 9 values per residue: 3 bond distances, 3 bond angles, and 3 dihedral torsional angles. The three bond angles and dihedrals are described in Table 1, and the three bond distances correspond to $N_i \rightarrow C\alpha_i$, $C\alpha_i \rightarrow C_i$, and $C_i \rightarrow N_{i+1}$ where $i$ denotes residue index. These 9 values enable a protein backbone to be *losslessly* converted from Cartesian to internal angle representation, and vice versa. To determine which subset of values to use to formulate proteins in our model, we take a set of experimentally profiled proteins and translate their coordinates from Cartesian to internal angles and distances and back, measuring the TM score between the initial and reconstructed structures. When excluding an angle or distance, we fix all corresponding values to the mean. The reconstruction TM scores of various combinations of values is illustrated in Figure S1. Of these 9 values, the three bond distances are the least important for reliably reconstructing a structure from Cartesian coordinates to the inter-residue representation and back; they can usually be replaced with constant average values without much impact on the recovered structure. In comparison, removing even two bond angles with relatively little variance ($\theta_2, \theta_3$) results in a large loss in reconstruction TM score (third bar). Removing all bond angles and retaining only dihedrals ($\phi, \psi, \omega$) results in only about half of proteins being able to be reconstructed (last bar). Thus, we model the three dihedrals and the three bond angles (second bar in Figure S1); this simplifies our prediction problem to use only periodic angular values (instead of a mixture of angular and real values) without a substantial loss in the accuracy of described structures. Future work might include additional modeling of these real-valued bond distances.

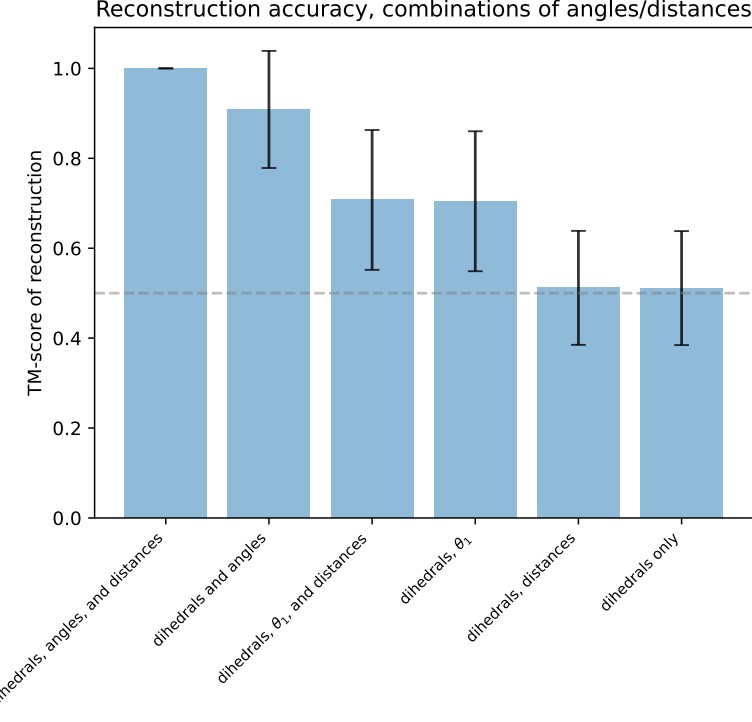

Figure S1: Various combinations of angles and distances and their ability to faithfully reconstruct protein backbones. A TM-score of 0.5 (dashed grey line) indicates the minimum similarity for two structures to be considered to have the same general shape. Error bars represent standard deviation in reconstruction TM scores. Using all bond angles, dihedral angles, and bond distances perfectly reconstructs Cartesian coordinates from internal angles (first column). The second column corresponds to the formulation used in the main text, where we model the 3 dihedrals and 3 bond angles, but keep the 3 bond distances fixed to average values. Other columns fix even more values to their respective means and result in reconstruction TM scores that are too low to be reliably useful.

One detail when converting between a $N$-residue set of Cartesian coordinates to a set of $N-1$ angles between consecutive residues is that the latter representation does not capture the first residue's information (as there is no prior residue to orient against). To solve this, we use a fixed set of coordinates to "seed" all generation of Cartesian coordinates, using the $N-1$ specified angles to build out from this fixed point. For all generations, this fixed point is extracted from the coordinates of the $N - C_\alpha - C$ atoms in the first residue on the N-terminus of the PDB structure 1CRN (Teeter, 1984). Doing so does not result in any meaningful reconstruction error in natural structures.

## A.2 EFFECT OF LENGTH ON STRUCTURE RECONSTRUCTION

One of the primary concerns of using an angle-based formulation is that small errors might propagate across many residues to culminate in a large difference in overall structure. To try and quantify the effect of this, we evaluate the "lossiness" of the representation itself, and the ability of the model to learn long-range angle dependencies.

We start by evaluating the accuracy of our representation itself over different structure lengths. To do this, we sample 5000 structures of varying length from the CATH dataset. For each, we compare the original 3D coordinate representation $x_c$ and the 3D coordinates obtained after converting the structure to angles and back to coordinates $\hat{x}_c$ using the TM score algorithm, i.e., $\text{TMscore}(x_c, \hat{x}_c)$. We find that longer structures exhibit greater TM score divergences when converted through our representation (Figure S2). However, even at our maximum considered structure length of 128 residues, structures still retain a reconstruction $\text{TMscore} \approx 0.9$, which is well above the accepted threshold of 0.5 denoting the same fold. This indicates that while our representation itself is slightly "lossy", the losses do not change the overall fold. Even when considering longer structures up to 512 residues in length, the reconstructed structures still share a TMscore similarity much greater than 0.5 (graph not shown), which suggests that our method could scale up to larger structures.

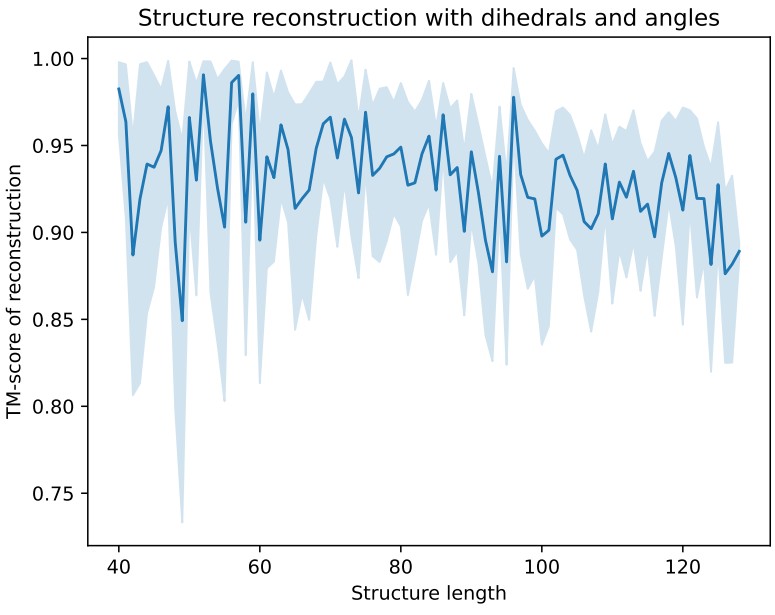

Figure S2: Faithfulness of reconstruction (via TMscore, y-axis) when using the 3 dihedrals and 3 angles described in Table 1 and keeping bond distances fixed to average values, evaluated across 5000 structures of varying length (x-axis). We observe a significant negative correlation between length and reconstruction TM score (Spearman's correlation $r = -0.18$, $p = 7.2 \times 10^{-39}$).

Next, we evaluate our model's ability to successfully reconstruct sequences of varying length. For each structure in our held-out test set ($n = 3040$), we add $t = 750$ timesteps of noise to that structure's angles (recall that we use a total of $T = 1000$ timesteps during training). This adds a significant amount of noise to corrupt the structure, while retaining a hint of the target true structure as a weak "guide" to what the model should ultimately denoise towards. We then apply our trained

model to these mostly-noised examples, running them for the requisite 750 iterations to fully denoise and reconstruct the angles. Afterwards, we assess reconstruction accuracy by taking the TMscore between the structure specified by the reconstructed angles, and the true structure specified by the ground truth angles. Comparing structures specified by reconstructed and true angles isolates the effects of model error, irrespective of any minor lossiness induced by the representation itself, which we investigated previously (Figure S2). Figure S3 illustrates the relationship between test set structure length and this reconstruction TM score. We find no significant correlation between length and reconstruction TM score (Spearman's correlation $r = -0.0024, p = 0.89$). FoldingDiff accurately reconstructs structures with a TM score of greater than 0.95 for 98% of test set, and achieves an average test reconstruction TM score of 0.988.

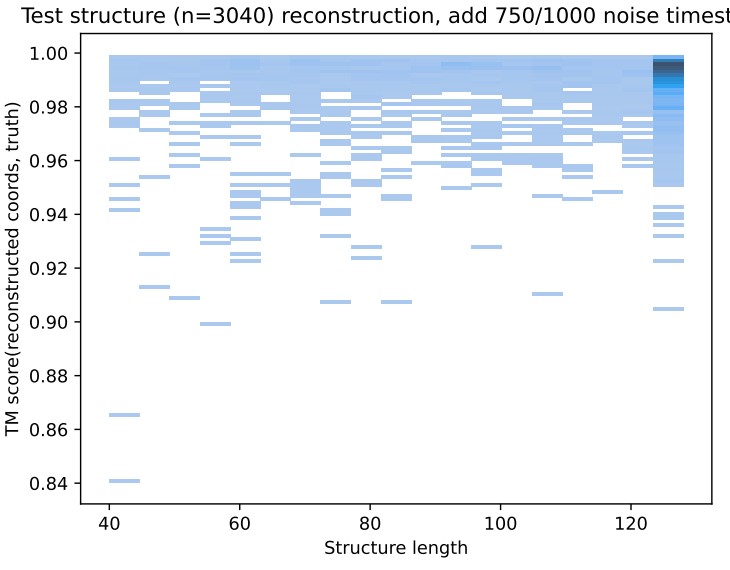

Figure S3: Test set structure reconstruction accuracy after injecting 750 timesteps of noise into angles. This is evaluated by taking the TM score between the structured specified by the reconstructed angles, and the structure specified by the ground truth angles. The x-axis denotes length of the test set structure, and each bar denotes the distribution of TM scores within that length.

All in all, these results indicate that (1) our representation itself is indeed lossier with longer structures, but not to a degree that would significantly impact the overall structures, and (2) our model is capable of learning robust relationships between angles regardless of structure length.

## B    ADDITIONAL NOTES ON SELF-CONSISTENCY TM SCORE

Our scTM evaluation pipeline is similar to previous evaluations done by Trippe et al. (2022) and Lee & Kim (2022), with the primary difference that we use OmegaFold (Wu et al., 2022) instead of AlphaFold (Jumper et al., 2021). OmegaFold is designed without reliance on multiple sequence alignments (MSAs), and performs similarly to AlphaFold while generalizing better to orphan proteins that may not have such evolutionary neighbors (Wu et al., 2022). Furthermore, given that prior works use AlphaFold without MSA information in their evaluation pipelines, OmegaFold appears to be a more appropriate method for scTM evaluation.

OmegaFold is run using default parameters (and `release1` weights). We also run AlphaFold without MSA input for benchmarking against Trippe et al. (2022). We provide a single sequence reformatted to mimic a "MSA" to the `colabfold` tool (Mirdita et al., 2022) with 15 recycling iterations. While the full AlphFold model runs 5 models and picks the best prediction, we use a singular model (`model1`) to reduce runtime.

Trippe et al. (2022) use ProteinMPNN (Dauparas et al., 2022) for inverse folding and generate 8 candidate sequences per structure, whereas Lee & Kim (2022) use ESM-IF1 (Hsu et al., 2022)

and generate 10 candidate sequences for each structure. We performed self-consistency TM score evaluation for both these methods, generating 8 candidate sequences using author-recommended temperature values ($T = 1.0$ for ESM-IF1, $T = 0.1$ for ProteinMPNN). We use OmegaFold to fold all amino acid sequences for this comparison. We found that ProteinMPNN in $C_\alpha$ mode (i.e., alpha-carbon mode) consistently yields much stronger scTM values (Tables S1, S2); we thus adopt ProteinMPNN for our primary results. While generating more candidate sequences leads to a higher scTM score (as there are more chances to encounter a successfully folded sequence), we conservatively choose to run 8 samples to be directly comparable to Trippe et al. (2022). We also use the same generation strategy as Trippe et al. (2022), generating 10 structures for each structure length $l \in [50, 128)$ – thus the only difference in our scTM analyses is the generated structures themselves.

## C  ABLATIONS AND BASELINES

### C.1  SUBSTITUTING INTERNAL ANGLE FORMULATION FOR CARTESIAN COORDINATES

We perform an "ablation" of our internal angle representation by replacing our framing of proteins as a series of inter-residue internal angles with a simple Cartesian representation of $C_\alpha$ coordinates $x \in \mathbb{R}^{N \times 3}$. Notably, this Cartesian representation is no longer rotation or shift invariant. We train a denoising diffusion model with this Cartesian representation, using the same variance schedule, transformer backbone architecture, and loss function, but sampling from a standard Gaussian and with all usages of our wrapping function $w$ removed. This represents the same modelling approach as our main diffusion model, with only our internal angle formulation removed.

To evaluate the quality of this Cartesian-based diffusion model's generated structures, we calculate the pairwise distances between all $C_\alpha$ atoms in its generated structures and compare these with distance matrices calculated for real proteins and for our internal angle diffusion model's generations. For a real protein, this produces a pattern that reveals close proximity between pairwise residues where the protein is folded inwards to produce a compact, coherent structure (Figure S4a). However, similarly visualizing the $C_\alpha$ pairwise distances in the Cartesian model's generated structures yields no significant proximity or patterns between any residues (Figure S4b). This suggests that the ablated Cartesian model cannot learn to generate meaningful structure, and instead generates a nondescript point cloud. Our internal angle model, on the other hand, produces a visualization that is very similar to that of real proteins (Figure S4c). Simply put, our model's performance drastically degrades when we change only how inputs are represented. This demonstrates the importance and effectiveness of our internal angle formulation.

### C.2  AUTOREGRESSIVE BASELINE FOR ANGLE GENERATION

As a baseline method for our generative diffusion model, we also implemented an autoregressive (AR) transformer $f_{\text{AR}}$ that predicts the *next* set of six angles in a backbone structure (i.e., the same angles used by FoldingDiff, described in Figure 1 and Table 1) given all *prior* angles.

Architecturally, this model consists of the same transformer backbone as used in FoldingDiff combined with the same regression head converting per-token embeddings to angle outputs, though it is trained using absolute positional embeddings rather than relative embeddings as this improved validation loss. The total length of the sequence is encoded using random Fourier feature embeddings, similarly to how time was encoded in FoldingDiff, and this embedding is similarly added to each position in the sequence of angles. The model is trained to predict the $i$-th set of six angles given all prior angles, using masking to hide the $i$-th angle and onwards. We use the same wrapped smooth L1 loss as our main FoldingDiff model to handle the fact that these angle predictions exist in the range $[-\pi, \pi]$; specifically: $L_w(x^{(i)}, f_{\text{AR}}(x^{(0,\dots,i-1)}))$ where superscripts indicate positional indexing. This approach is conceptually similar to causal language modeling (Radford et al., 2019), with the important difference that the inputs and outputs are continuous values, rather than (probabilities over) discrete tokens.

This model is trained using the same data set and data splits as our main FoldingDiff model with the same preprocessing and normalization. We train $f_{\text{AR}}$ using the AdamW optimizer with weight decay set to 0.01. We use a batch size of 256 over 10,000 epochs, linearly scaling the learning rate from 0 to $5 \times 10^{-5}$ over the first 1,000 epochs, and back to 0 over the remaining 9,000 epochs.

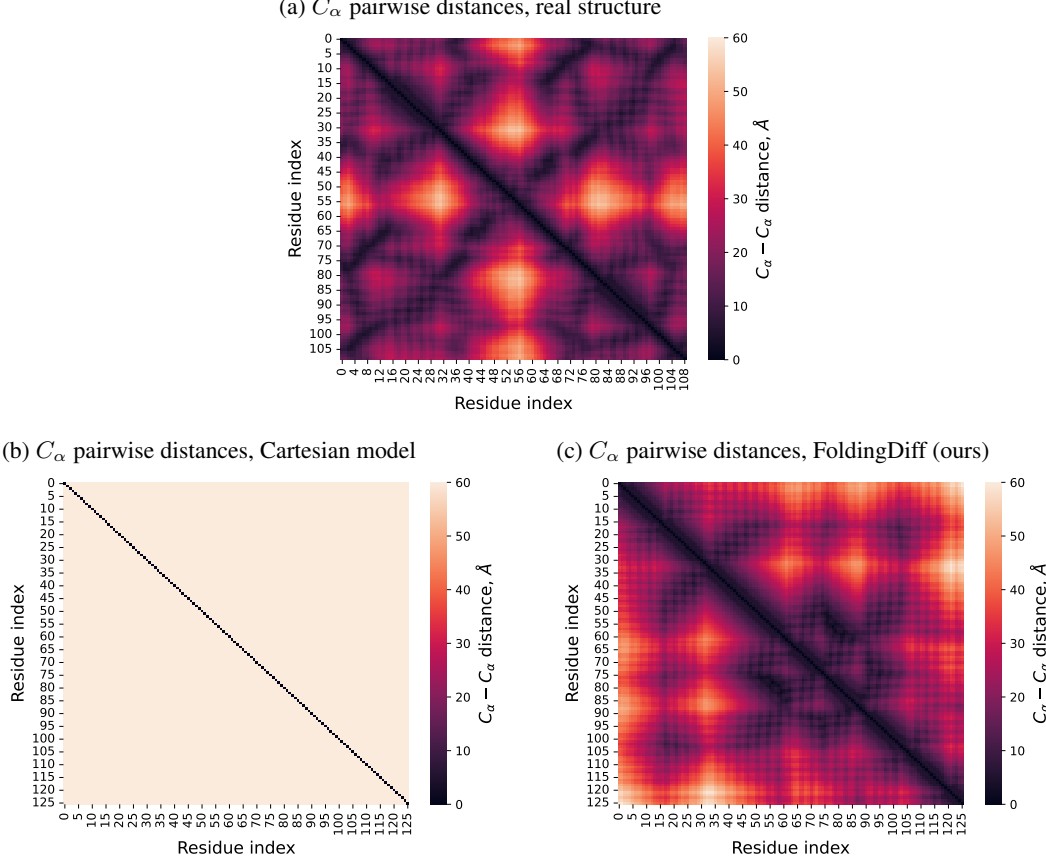

Figure S4: Pairwise distances between all $C_\alpha$ atoms in various protein backbone structures, all of similar length. All panels use the same color scale. S4a illustrates a set of distances for a real protein structure; note the visual patterns that correspond to various secondary structures and potential contacts and interactions between residues. S4b shows these distances for a structure generated by an ablated model that replaces our proposed internal angle representation with Cartesian coordinates, which results in no coherent structural generation. For comparison, our FoldingDiff model produces structures that compactly fold to create many potential contacts, just as real proteins do (S4c).

To generate structures from $f_{AR}$, we "seed" the autoregressive model with 4 sets of 6 angles taken from the corresponding first 4 angle sets in a randomly-chosen naturally occurring protein structure. This serves as a random, but biologically realistic, "prompt" for the model to begin generation. We then supply a fixed length $l$ and repeatedly run $f_{AR}$ to obtain the next $i$-th set of angles, appending each prediction to the existing $i - 1$ values in order to predict the $i + 1$ set of angles. We repeat this until we reach our desired structure length.

We use the above procedure to generate 10 structures for each structure length $l \in [50, 128)$ each with a different set of seed angles. Examples of generated structures of varying lengths are illustrated in Figure S5. All structures generated by this autoregressive approach consist of one singular $\alpha$ helix. This is confirmed by running P-SEA on the generated structures (Figure S6a). Besides being an obvious sign of modal collapse, these extremely long $\alpha$ helices are also not commonly observed in nature (Qin et al., 2013).

Such behavior where autoregressive models generate a single looped pattern endlessly has been observed in language models as well, where it is often circumvented using a temperature parameter to inject randomness. In our continuous regime, we try to do something similar by adding small amounts of noise to each angle during generation, but are unable to meaningfully deviate from the aforementioned modal collapse to endless $\alpha$ helices.

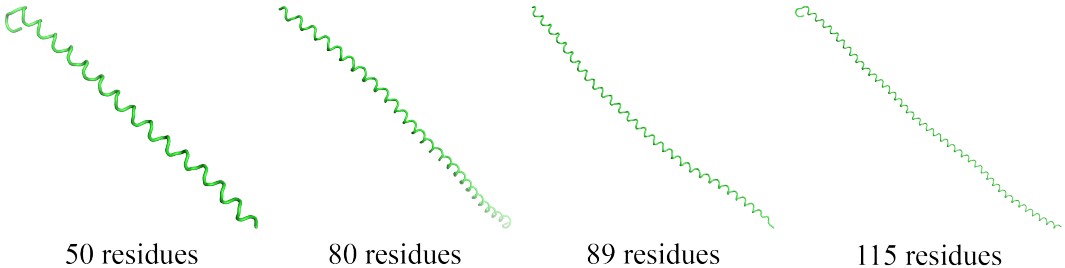

| 50 residues | 80 residues | 89 residues | 115 residues |

Figure S5: Structures generated using an autoregressive (AR) baseline approach trained on the same angle-based formulation we propose. This approach predicts the next set of angles given all prior angles, and can be thus used to iteratively generate structures in an autoregressive fashion (see Appendix C.2 for additional details). However, the structures generated this way are all straight $\alpha$ helices, regardless of initial "prompt" angles (see Figure S6a). This complete lack of diversity and meaningful complexity indicates that while this AR model can produce technically "correct" structures, it cannot be used for generative modeling to any meaningful capacity. Figure 6 analogously illustrates FoldingDiff's generations, which are structurally much more diverse.

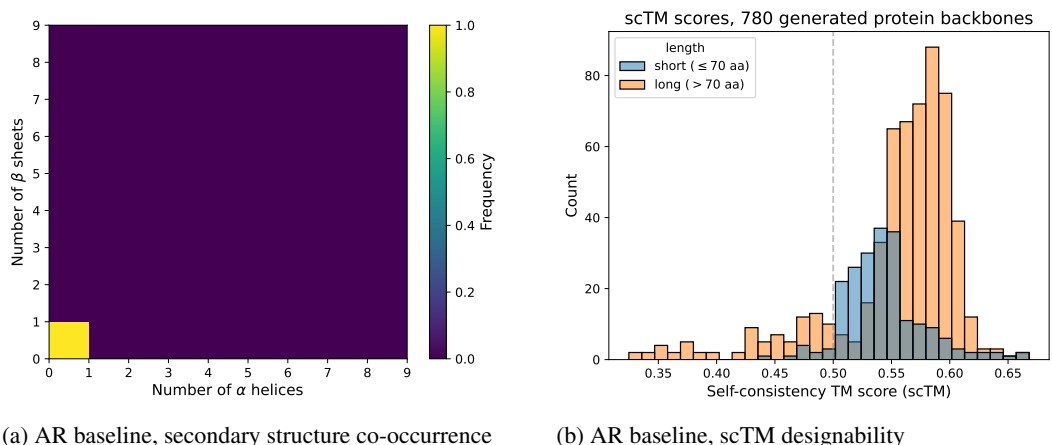

(a) AR baseline, secondary structure co-occurrence  (b) AR baseline, scTM designability

Figure S6: Secondary structure elements (a) and designability (b) for structures generated by the autoregressive baseline described in Appendix C.2. We observe that using P-SEA to annotate these generated structures detects exclusively *singular* $\alpha$ helices, and no $\beta$ sheets (S6a). This quantifies the observations in Figure S5 that the AR model has "collapsed" into repeatedly generating these coils. We find that these helices exhibit greater designability via scTM scores (computed using ProteinMPNN and OmegaFold) (S6b), with 693 of the 780 structures having scTM $\geq 0.5$, though this increase is not meaningful due to the utter lack of diversity in generated sequences.

We evaluate the AR model's generated structures' scTM designability using ProteinMPNN (CA-only mode) and OmegaFold. We find that 693 of the 780 generated structures are designable, leading to an overall designability ratio of 0.89. (Figure S6b). Importantly however, this gain in designability is meaningless, as singular endless coils are biologically neither useful nor novel.

## C.3 BASELINES CONTEXTUALIZING SCTM SCORES

To contextualize FoldingDiff's scTM scores (Figure 5a), we implement a naive angle generation baseline. We take our test dataset, and concatenate all examples into a matrix of $\hat{x} \in [-\pi, \pi)^{\hat{N} \times 6}$, where $\hat{N}$ denotes the total number of angle sets in our test dataset, aggregating across all individual chains. To generate a backbone structure of length $l$, we simply sample $l$ indices from $U(0, \hat{N})$. This creates a chain that perfectly matches the natural distribution of protein internal angles, while also perfectly reproducing the pairwise correlations, i.e., of dihedrals in a Ramachandran plot, but

critically loses the correct *ordering* of these angles. We randomly generate 780 such structures (10 samples for each integer value of $l \in [50, 128)$). This is the same distribution of lengths as the generated set in our main analysis. For each of these, we perform scTM evaluation with ProteinMPNN (CA-only mode) and OmegaFold. The distribution of scTM scores for these randomly-sampled structures compared to that of FoldingDiff's generated backbones is shown in Figure S7. We observe that this random protein generation method produces significantly poorer scTM scores than FoldingDiff ($p = 1.6 \times 10^{-121}$, Mann-Whitney test). In fact, not a single structure generated this way is designable. This suggests that our model is not simply learning the overall distribution of angles, but is learning *orderings and arrangements* of angles that comprise folded protein structures.

Figure S7: Distribution of scTM scores for our generated structures (orange), compared to scTM scores for structures created by randomly shuffling naturally-occurring internal angles (blue). The randomly sampled angles result in no designable structures, despite perfectly capturing the overall distribution and pairwise relations between angles. This suggests our method correctly learns the spatial ordering of angles that folds a valid structure. We additionally take a set of 780 experimental structures and pass them through our scTM pipeline to evaluate the fragility of this pipeline itself. We find that 87% of natural proteins (green) are designable; this forms a "soft" upper bound for what would be achievable by sampling from the true data distribution.

We take the structures specified by these randomly sampled angles and analyze them for secondary structures using P-SEA, as we did for Figure 4. The resulting 2D histogram is illustrated in Figure S8, and represents a completely different distribution of secondary structures compared to natural structures, or that of FoldingDiff's generations. This further suggests that this random angle baseline cannot generate natural-appearing structures. It is clear that scTM scores and secondary structure analyses cannot be satisfied using this trivial solution.

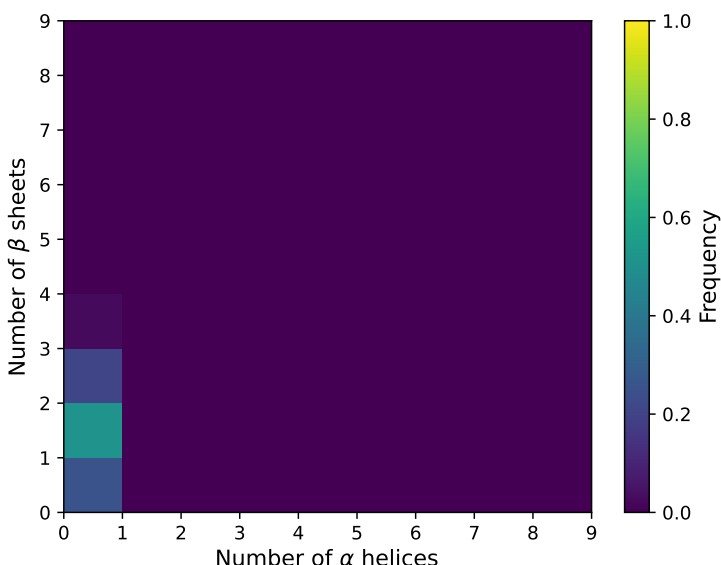

Figure S8: Secondary structures present in structures generated by randomly shuffling naturally-occurring angles, as described in Appendix C.3. This random baseline produces a few hits to $\beta$ sheets by chance, but notably lacks the $\alpha$ helices present in both natural structures and FoldingDiff's generations (Figures 4, S11).

# D  ADDITIONAL SUPPLEMENTARY FIGURES AND TABLES

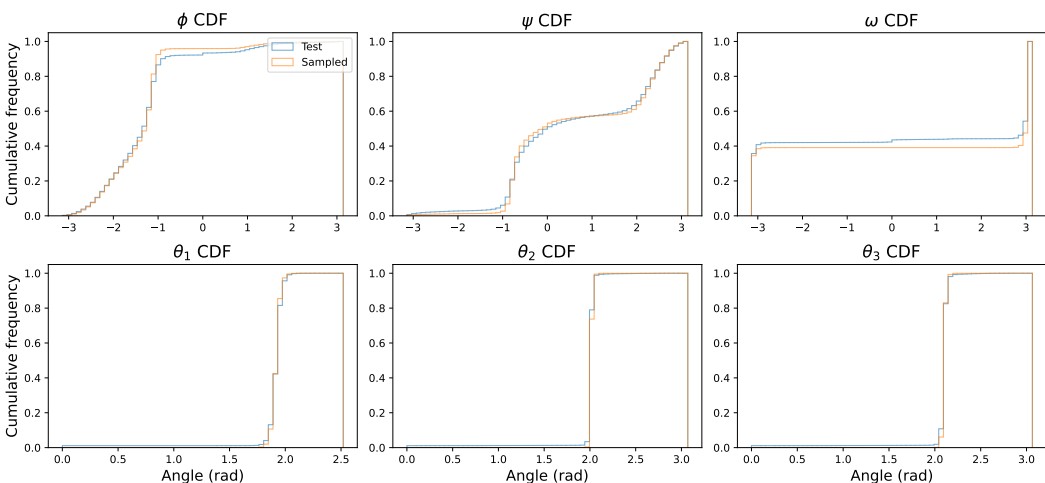

Figure S9: Comparison of the cumulative distribution functions (CDF) of angular values in test set and in generated samples. Top row shows dihedral angles (torsional angles involving 4 atoms), and bottom row shows bond angles (involving 3 atoms). Figure 2 shows the histogram distributions corresponding to these CDFs.

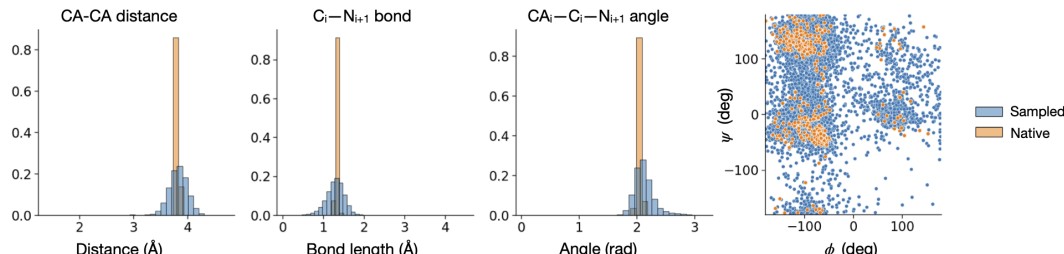

Figure S10: Figure 1B from Anand & Achim (2022), reproduced with permission for ease of reference. Illustrated $C\alpha_i - C_i - N_{i+1}$ bond angle (third plot from the left) corresponds to $\theta_2$ in our formulation. Sampled angles in the work of Anand & Achim (2022) exhibit a much larger spread than the natural distribution of angles, whereas our work matches much more tightly (Figures 2, S9).

Table S1: Self-consistency TM scores (using ProteinMPNN and OmegaFold) across replicates. Each run generates 10 different structures for each length in $l \in [50, 128]$, resulting in 780 total generated structures, and is started from a different random seed, using the same pre-trained model as in the primary text and Appendix C.3. Self-consistency TM scores are computed using ProteinMPNN and OmegaFold, as described in the primary text. Short structures are defined as having 70 residues or fewer ($n = 210$); long structures are defined as having more than 70 residues ($n = 570$). Values from our main text are reproduced in the last row for ease of reference. Each generation run produces a consistently high number of designable structures, and also contains a realistic mixture of secondary structure elements (Figure S11).

| Seed | Designable | Designable, short ($n = 210$) | Designable, long ($n = 570$) |
|---|---|---|---|
| 1 | 173 | 72 | 101 |
| 2 | 154 | 75 | 79 |
| 3 | 185 | 90 | 95 |
| 4 | 182 | 86 | 96 |
| 5 | 187 | 76 | 111 |
| 7344 (main text) | 177 | 80 | 97 |

Table S2: Self-consistency TM scores calculated using ESM-IF1 to perform inverse folding, rather than ProteinMPNN. These analyze the same FoldingDiff generations as Table S1, and are likewise folded with OmegaFold. ESM-IF1 consistently results in much lower scTM scores.

| Seed | Designable | Designable, short ($n = 210$) | Designable, long ($n = 570$) |
|---|---|---|---|
| 1 | 117 | 61 | 56 |
| 2 | 105 | 64 | 41 |
| 3 | 122 | 76 | 46 |
| 4 | 115 | 72 | 43 |
| 5 | 126 | 67 | 59 |
| 7344 (main text) | 111 | 57 | 54 |

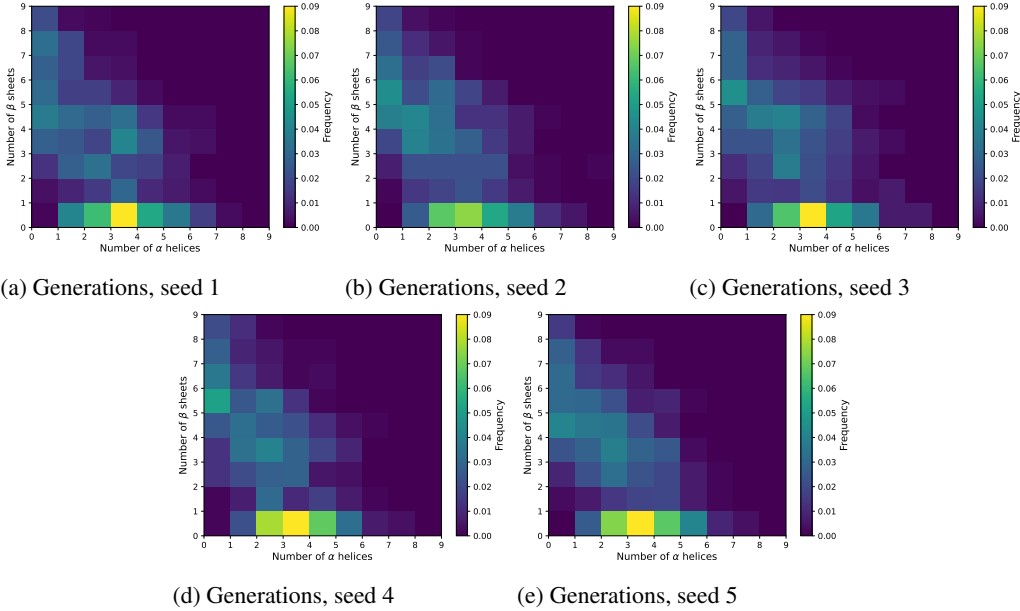

(a) Generations, seed 1     (b) Generations, seed 2     (c) Generations, seed 3

(d) Generations, seed 4     (e) Generations, seed 5

Figure S11: For each of the 5 replicates shown in Table S1, we use P-SEA to annotate secondary structures. Each run's generations contain a mixture of $\alpha$ helices (x-axis) and $\beta$ sheets (y-axis) that is comparable to natural structures (Figure 4a). FoldingDiff generates reasonable, complex structures consistently.

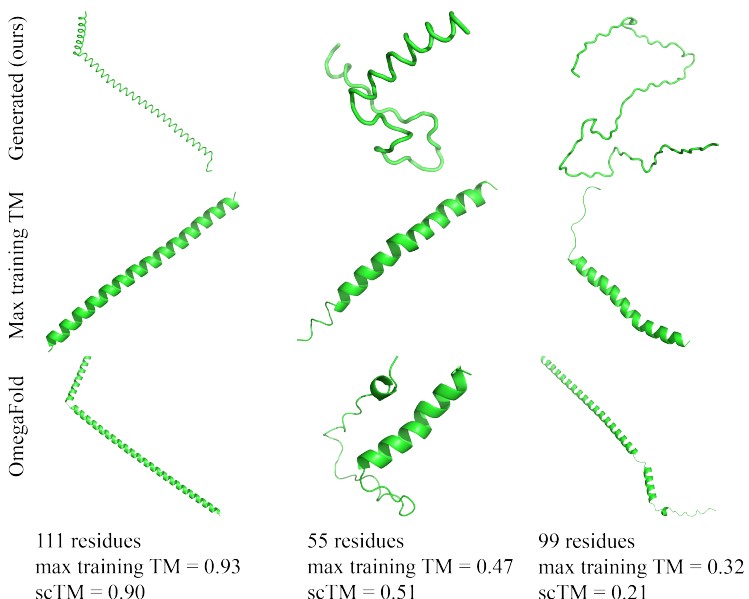

Figure S12: Generated structures representing the full range of designability (scTM) and training similarity scores. The top row indicates the original generated structure, the middle row shows the training structure with the highest TM score, and the bottom row indicates the structure predicted by OmegaFold based on residues predicted to produce our generated structure by ProteinMPNN. The first column shows a structure with high designability and high training similarity. The second column shows a structure with designability and training similarity close to 0.5. The third column shows a generated structure that is very different from any training chain, but is also not designable.

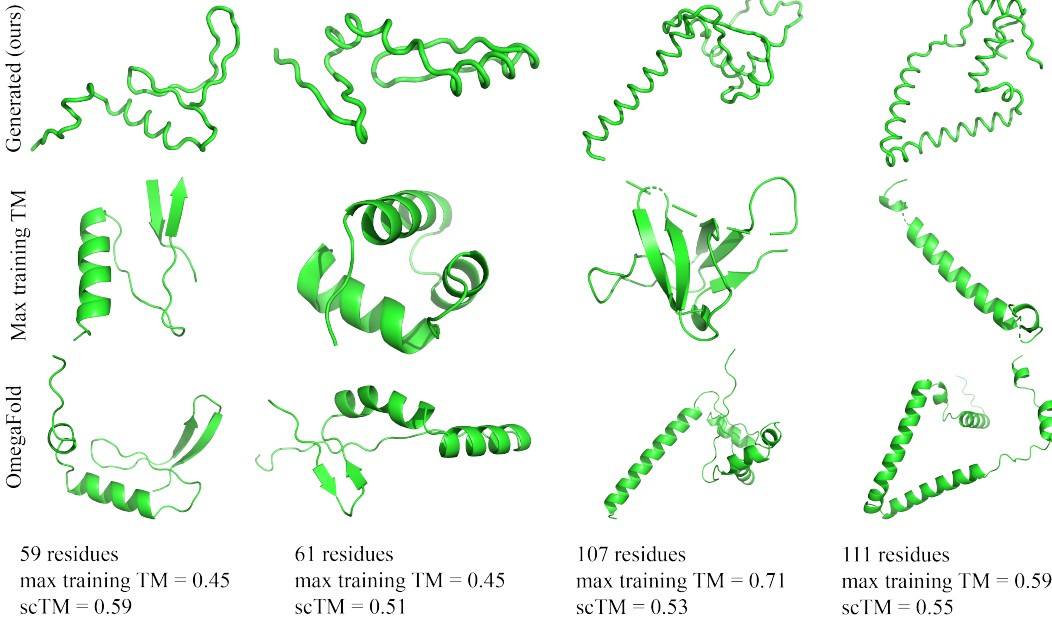

Figure S13: Structures from Figure 6 illustrated with the training example with the highest TM score (most similar). Figure rows are arranged as in Figure S12. Our generated structures are visually quite different compared to the best training set match – in almost every example, our generated structure contains a completely different arrangement of secondary structure elements than the closest training structure, indicating that they may be more distinct than TM scores alone might suggest.

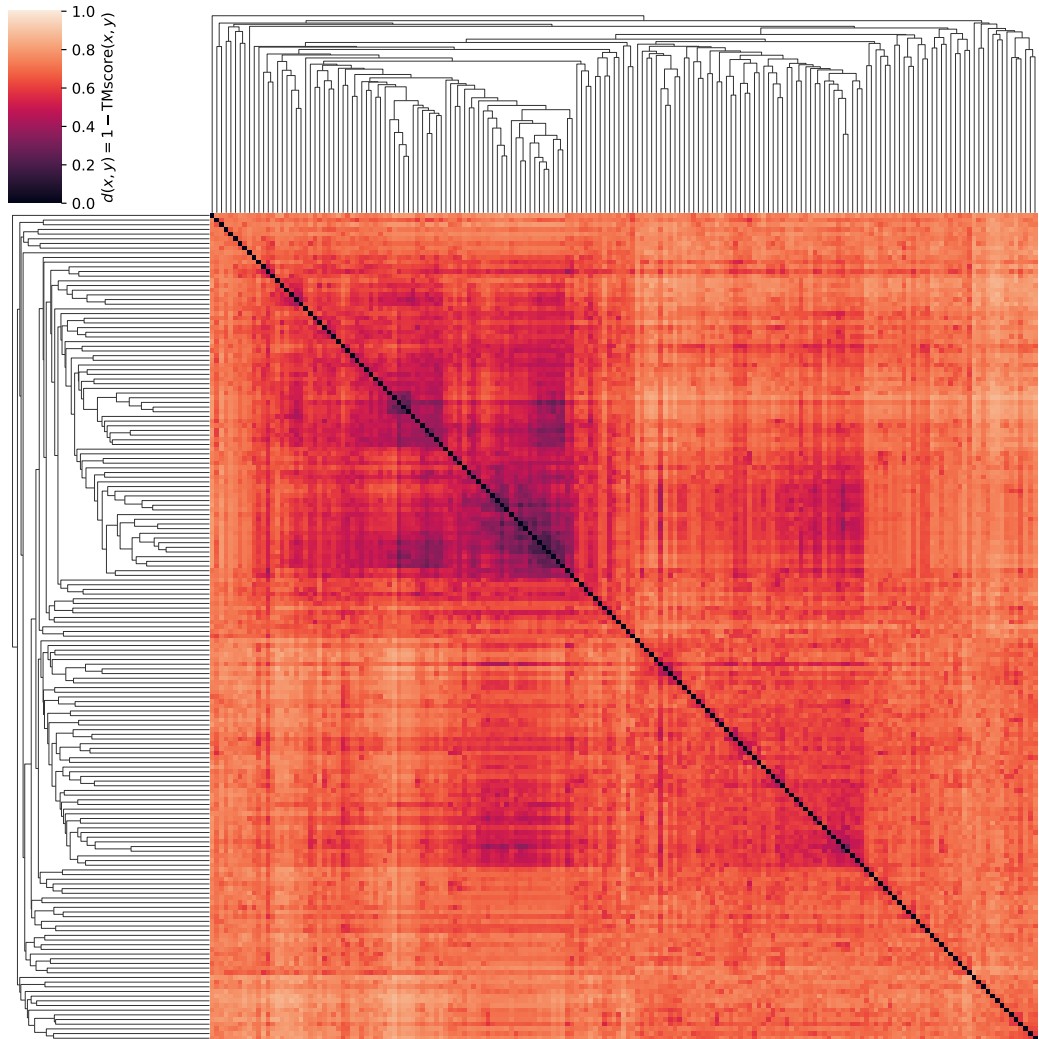

Figure S14: Clustering of our $n = 177$ "designable" generated backbones with scTM scores $\geq$ 0.5. We use the average distance metric to perform hierarchical clustering on the pairwise distance matrix $d(x, y) = 1 - \mathrm{TMscore}(x, y)$. Dark values corresponding to 0 (or conversely, a TM score of 1) indicate (nearly) identical structures. While there are some loosely related groups of structures, we do not observe clearly delineated groups. This indicates that the designable backbones we generate are diverse and represent a wide range of potential structures. For comparison to naturally-occurring structures and prior works, see Figures S15 and S16.

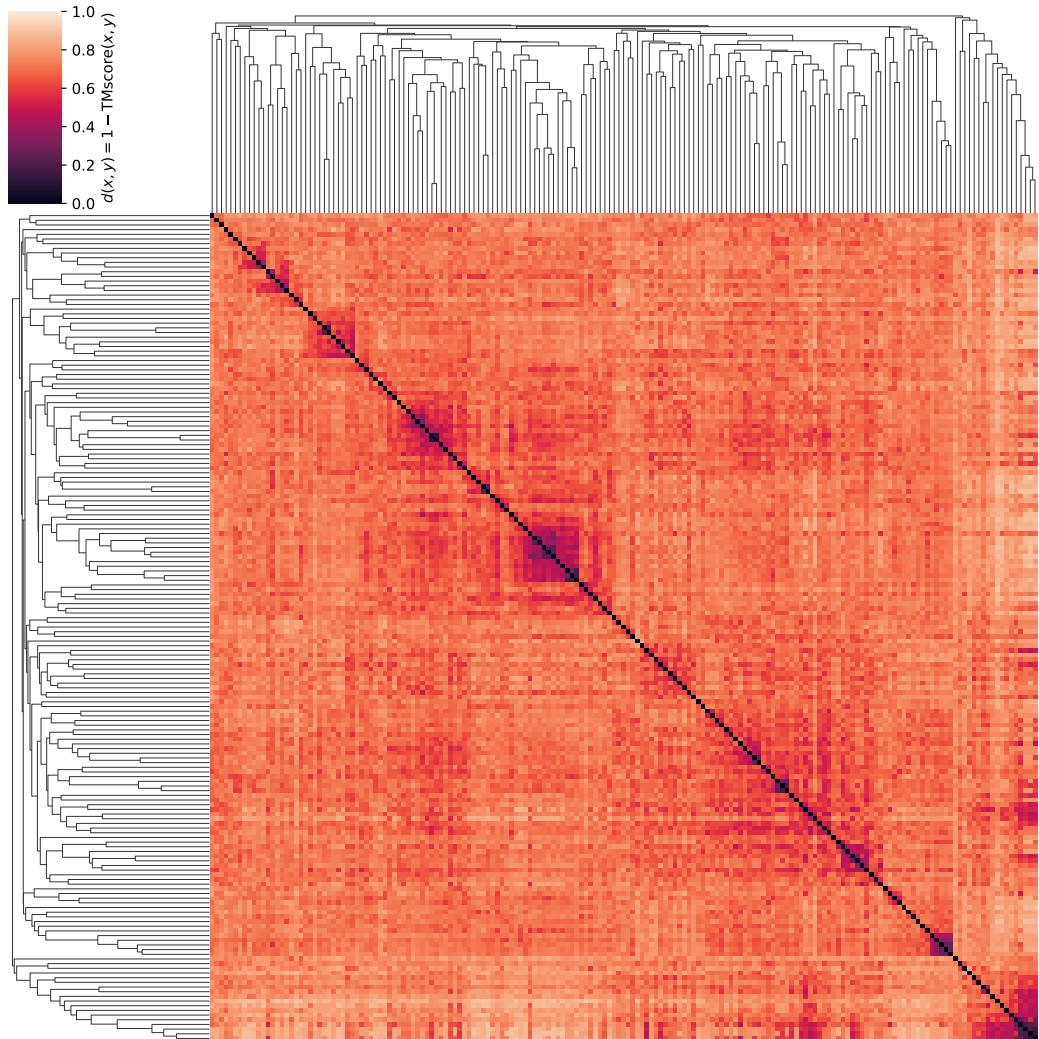

Figure S15: To provide additional context for the degree of diversity that is reasonable to expect from our generated sequences, we similarly perform clustering on a set of 177 randomly chosen naturally-occurring CATH structures between 50 and 128 residues in length. We find that natural sequences tend to cluster much more similarly to our designable sequences (Figure S14) compared to that of prior works (Figure S16).

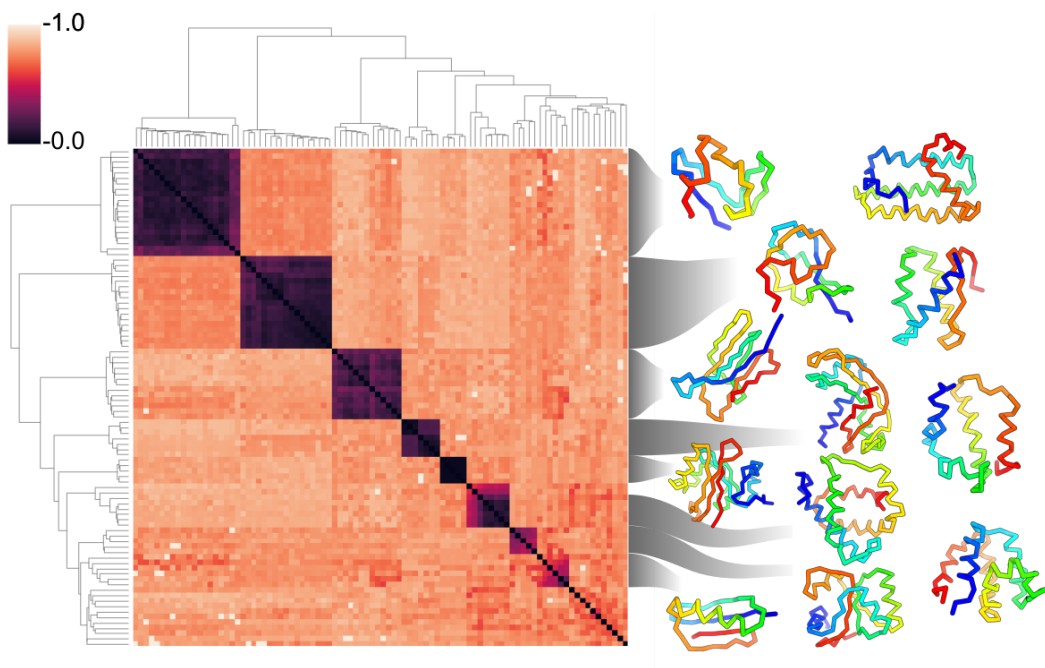

Figure S16: Figure from Trippe et al. (2022), reproduced with permission for ease of reference. These authors similarly cluster unconditionally generated backbones with scTM $\geq 0.5$ using $1 -$ TMscore$(x, y)$ as a distance metric. Compared to our identical evaluation, illustrated in Figure S14, we notice that this clustering has a few dark blocks of nearly 0 distance, or a TM score of nearly 1. This suggests that among these designable backbones, many are actually minor variants of a core structure; in actuality, though this work claims to produce 92 designable structures, there seem to be fewer unique structures in this set due to many being near-duplicates.

