# OpenReview forum: "Protein structure generation via folding diffusion"
_ICLR.cc/2023/Conference — Submitted to ICLR 2023_

### Official Review · Reviewer_84e9 · 2022-10-23

**Confidence:** 4
**Correctness:** 4
**Technical Novelty And Significance:** 3
**Empirical Novelty And Significance:** 4
**Recommendation:** 8

**Clarity, Quality, Novelty And Reproducibility:**

Clarity:
The paper is written in a very clear and accessible way.

Quality:
Idea is well motivated and shows convincing performance. Benchmarking is adequate.

Originality:
Diffusing angles is an original idea.

**Strength And Weaknesses:**

Strengths
- Novel approach for modeling 3D structure
- Convincing performance

Weaknesses
- Generated proteins are very short (not a problem within the claimed scope, the benchmarking is done well, but will be interesting to see how it scales - with increasing length, there could be an increase in collisions?)

Questions
- Does the presented approach really mirror native folding, as claimed in the abstract? I.e. when tracking the diffusion process through time does it recapitulate a folding process? I understand that the angle parametrization is closer to the native process, but that doesn't necessarily mean them model will learn to follow this process when denoising (clashes etc.)
- Could the authors elaborate on why to use ESM-IF1 over alternatives such as ProteinMPNN, and why to generate exactly 8 sequences?

**Summary Of The Paper:**

The paper proposes a denoising diffusion probabilistic model for generating protein backbone structures. To overcome the complication of equivariance when working with 3D structures, the approach directly models the angles that describe the orientation of residues relative to their neighbors. Results show that the model generates diverse structures that recapitulate the test distribution to a large extent

**Summary Of The Review:**

The paper introduces a novel parametrization of the protein structure generation problem by modeling angles instead of coordinates. Results convincingly show that it has some advantages over directly generating coordinates using equivariant model architectures.

---

> ### Author Response · Authors · 2022-11-15
> **Response to Reviewer 84e9**
>
> We thank the reviewer for their insightful comments and excitement for our work. Please see our general comments above, as well as responses to your specific questions below.
>
> ### Our approach and its relation to native folding
> We clarify that our approach does not follow any thermodynamic laws that would govern protein folding, nor does it have any restraints that prevent clashes or collisions. In other words, we do not explicitly perform “folding” as an energetic simulation. Our claims are merely that proteins fold by updating their angles towards more stable states, and that our diffusion process is inspired by this high-level intuition (without considering the actual underlying thermodynamics).
>
> ### Use of ESM-IF1
> We use ESM-IF1 because we empirically observed better performance than with ProteinMPNN; ProteinSGM (Lee et al.) also uses ESM-IF1 in their analyses. We generate 8 different sequences for each structure as this is done by both Lee et al. and Trippe et al.; this allows for more direct comparison of designability ratios. Ideally, we would rerun our analyses with comparisons to structures generated from these works, but neither of these works releases code or generated structures.
>
> ### Additional updates
> We have added several new analyses to better detail our approach’s limitations and advantages. Briefly, we have:
>
> * Added an autoregressive method as an additional baseline for generating series of angles; we find that this baseline invariably collapses into the failure mode of generating endless alpha helices and is thus not useful from a generative standpoint (new Appendix C.2, Figures S5, S6). Notably, our method is not subject to this modal collapse and generates diverse, complex structures.
> * Better disentangled our model’s dependency on length (new Appendix A.2, Figures S2, S3). We find that our angular representation itself tends to introduce more noise as sequences grow longer, though never to a point that is meaningfully harmful to the overall structure. We find that our model itself is able to effectively learn angles and their dependencies regardless of structure length.
> * Performed additional sampling runs from our model to show that our results are not the result of a lucky run (Table S1). In fact, the run we show in our main text has the lowest performance of the seeds we have additionally evaluated. These new generation runs also contain rich mixtures of alpha and beta secondary structures (new Figure S11).
>
> We hope these clarifications address the reviewer’s questions, and that these new analyses lend further strength to the reviewer’s support of the manuscript.
>
> ### References
> * Lee, Jin Sub, and Philip M. Kim. "ProteinSGM: Score-based generative modeling for de novo protein design." bioRxiv (2022).
> * Trippe, Brian L., et al. "Diffusion probabilistic modeling of protein backbones in 3D for the motif-scaffolding problem." arXiv preprint arXiv:2206.04119 (2022).

---

> > ### Author Response · Authors · 2022-11-18
> > **Response to Reviewer 84e9, re ProteinMPNN + AlphaFold**
> >
> > We thank the reviewer for bringing our attention back to ProteinMPNN. We originally ran ProteinMPNN in full atom mode, which generated only glycine residues, but upon revisiting our evaluation pipeline, we found that using ProteinMPNN’s CA model provides significantly improved performance compared to ESM-IF1: for our main results, designability is improved from 111/780 to 177/780 designable sequences for inverse folding with ESM-IF1 vs. ProteinMPNN, respectively. In light of these new results, we have updated our main text with results that use ProteinMPNN. All text, Table S1, and Figures 5, 6, S6b, S7, S12, S13, S14, S15 have been updated accordingly. In addition, we have added new text that details the difference between ESM-IF1 and ProteinMPNN (Appendix B) as well as a new Table S2 that retains our original numbers for multiple replicates with ESM-IF1.
> >
> > In a similar spirit, we have also run AlphaFold on ProteinMPNN’s sequences corresponding to the generated structures presented in our main text. This brings our scTM pipeline in full alignment with that used by Trippe et al. Using this evaluation pipeline, we find a designability of 163/780, which significantly exceeds this prior work’s claim of 92/780. This improvement in designability is significant for both short structures up to 70 residues, as well as longer structures up to 128 residues.

---

### Official Review · Reviewer_DFe6 · 2022-10-24

**Confidence:** 3
**Correctness:** 3
**Technical Novelty And Significance:** 2
**Empirical Novelty And Significance:** 1
**Recommendation:** 3

**Clarity, Quality, Novelty And Reproducibility:**

- Paper is reasonable clear, and touches on a range of analyses.
- angles for structure prediction is already presented in Chowdhury 2021 and AlQuraishi 2019, but this paper fails to cite these works.

**Strength And Weaknesses:**

Strengths:
- Many qualitative measure of model performance, ranging from rama plots to scTM scores
- Diffusing on angles is relatively well motivated, and interesting.

Weaknesses:
- Like many other works on protein structure generation, this paper does not solve the problem that good metrics are hard. Therefore, it's hard to compare how much better this model does compared to previous works. My suggestion is to produce a set of comprehensive samples, and compare them to other techniques that way.
- Many metrics selected are claims about overall statistics, not individual proteins. The few samples shown are underwelming
- The scTM score provided is underwhelming as well, having most scores significantly lower than 0.5. For well generated proteins, I would expect >0.8 to be reasonable. RMSD might be a more interpretable metric to display also - for which we're looking for sub 3 A self consistency.
- The model seems unable to generate beta sheets, which require the model to better understand the global structure of the protein.
- Why does the generated samples not show any secondary structure through pymol? That usually indicates to me that the proteins generated have significant deficiencies in stereochemical properties.

**Summary Of The Paper:**

The authors show that featurizing a protein via angles allows generation of protein structures via diffusion. They provide a technique for diffusing in angular space, and show that it's able to generate interesting looking protein structures. The authors provide many qualitative metrics to compare their model with recent structure generation techniques. The introduced technique seems to improve on angular characteristics of the generated proteins, though it does not seem to generate compact and interesting backbones, resulting in mostly alpha helices.

**Summary Of The Review:**

The work itself is reasonably solid, though the generations do not seem signifcantly better than existing works. Additionally, the paper does not solve the problem that structure generation works are hard to compare, which makes it difficult to know the impact of this work. The generated structures are underwhelming.

---

> ### Author Response · Authors · 2022-11-15
> **Response to Reviewer DFe6 (part 1/2)**
>
> We appreciate the reviewer’s feedback and clear critiques of our method and approach. We have responded to some general concerns regarding comparisons, metrics, and limitations in the general comment; please see below for responses to your specific concerns.
> ### Good metrics are hard
> We agree that good metrics are invaluable to evaluating generative works and yet hard to define; please see our general comments regarding this above. Our work uses a similar set of evaluations as recently disclosed generative works for protein structure (e.g., self-consistency TMscore) so as to be comparable as possible without access to these prior works’ code. Additional metrics like an FID analog for proteins would be invaluable to the field in general, but are not particularly meaningful when existing works have no code for verifiable benchmarking.
>
> To the suggestion of producing a set of comprehensive samples, this is indeed how we have evaluated our model in our work – by generating a large set of structures and holistically evaluating these structures across multiple metrics (Figures 2-5; Figure S4, S9, S11, S14). We believe evaluation of multiple samples across multiple metrics is critical – given that with a single metric, in isolation, it is difficult to assess the quality of generated structures. We have additionally performed new analysis that repeats generation across 5 additional random seeds, and show that our improvements to designability are consistent across all tested generation runs (new Table S1, reproduced below), and that these runs all contain realistic combinations of diverse secondary structure elements (new Figure S11).
>
> | Seed | Designable | Designable, short (n=210) | Designable, long (n=570) |
> | --- | --- | --- | --- |
> | 1 | 173 | 72 | 101 |
> | 2 | 154 | 75 | 79 |
> | 3 | 185 | 90 | 95 |
> | 4 | 182 | 86 | 96 |
> | 5 | 187 | 76 | 111 |
> | 7344 (main text) | 177 | 80 | 97 |
>
> ### Focus on overall vs. individual generated proteins
> Our paper aims to demonstrate that an angle-based formulation, as a new method, is a viable approach for protein structure generation, not that we can produce individual instances of designable proteins. With this goal in mind, we focus our evaluation on the overall distribution of proteins generated, because the goal of this work is to show that our formulation allows for designing more novel and designable proteins on average.
>
> To complement this overall analysis, we also provide representative examples to give readers a sense of the range of structures they can expect from our model. We caution against individually assessing these examples. Not only is this practice subjective, but our model also only generates structures up to 128 residues, so the expectation that these proteins look like “natural” proteins of unrestricted length is unrealistic – shorter proteins have limited opportunity to exhibit global complexity. Showing more structures, as the reviewer suggests, will not solve the issue that individual evaluation of structures is qualitative, subjective, and biased.
>
> Together, these analyses provide both broad, quantitative comparisons across large sets of generated structures as well as specific examples. We stress these considerations because our work is the first step at establishing a new angle-based method for protein generation; it does not claim to solve the problem of generating novel proteins in general.
>
> ### Concerns regarding performance
> The reviewer is correct in pointing out that most of the scTM scores are lower than 0.5. However, we point out that the threshold the reviewer sets for well-designable proteins (>0.8) is beyond even that of experimentally determined structures (see following paragraph). We also respectfully disagree that our model needs to perform as well as this experimental upper-bound to be a meaningful computational contribution to the field.
>
> We have added new analyses to establish an upper bound for scTM scores by taking native protein structures and passing them through our scTM pipeline. Even for these experimentally determined structures, only 80% have an scTM above 0.5, and only 43% have an scTM above 0.8 (updated Figure S7). Thus, it is unrealistic to expect a generative method to exceed scTM > 0.8.

---

> > ### Author Response · Authors · 2022-11-15
> > **Response to Reviewer DFe6 (part 2/2)**
> >
> > Current methods for protein structure generation fall substantially short of this upper bound. For example, Trippe et al. reports a lower scTM designability ratio than ours. Naive strategies like randomly sampling angles from the native distribution produces no designable structures whatsoever (Appendix C.3). As noted in our text, ProteinSGM (Lee et al.) reports a higher scTM ratio, but this is measured after using Rosetta to produce a foldable structure – hence the scTM score does not directly reflect their generative process. ProteinSGM’s structures also have extremely high similarity to their training set, suggesting that their higher scTM scores may come at the cost of novelty. Indeed, to date and to our knowledge, no prior work has demonstrated both high designability and high novelty from training structures.
> >
> > We have additionally added a new baseline model to further illustrate how challenging it is to generate protein structures. We train an autoregressive model to predict the next set of angles given all prior angles. We find that this model fails to generate any meaningful structures, instead generating endless alpha helices (new Appendix C.2, Figures S5, S6). FoldingDiff does not exhibit this modal collapse. We hope that this provides additional evidence that FoldingDiff provides advances on a fundamentally challenging problem.
> >
> > Thus, while we agree that many of the structures generated by FoldingDiff have a relatively low scTM, our method still closes the gap between computational methods for generative design and experimentally determined structures. Generating designable protein structures from scratch is a fundamentally hard problem, and although our method has limitations, it still reflects an advance towards this goal.
> >
> > ### Beta sheet generation
> > It is true that among designable proteins with scTM above 0.5, there are almost no beta sheets. However, among all generations, we find a similar occurrence of beta sheets compared to natural structures (Figure 4, new Figure S11). We agree that our model could better learn global structural motifs; however, we also believe that this avenue for future work does not detract from the value that this work presently has to the broader research community.
> >
> > ### PyMol visualizations
> > PyMol visualizations are not a substitute for explicit secondary structure annotations. These annotations are done in Figure 4 and new Figure S11, which clearly illustrates that our generated structures without refinement have consistent, detectable secondary structures.
> >
> > Even experimentally validated backbones can lack automatic secondary structures when visualized through PyMol. For example, Figure 1 shows an experimental structure in our training dataset, reconstructed from our angular representation (which fixes bond distances to average lengths), but is nonetheless nearly identical to the original structure. It is possible that PyMol is sensitive to small variations in bond distances, and is therefore not producing secondary structure cartoons. This is, again, why we caution against visually evaluating individual structures and run explicit secondary structure analyses as discussed above.
> >
> > ### References to angular approaches to structure prediction
> > As suggested, we have updated our text to reference additional non-generative works that use an angular representation to predict protein structure from protein sequence (Gao et al. 2017, AlQuraishi 2019, Chowdhury et al. 2021).
> >
> > ### References
> > * Lee, Jin Sub, and Philip M. Kim. "ProteinSGM: Score-based generative modeling for de novo protein design." bioRxiv (2022).
> > * Trippe, Brian L., et al. "Diffusion probabilistic modeling of protein backbones in 3D for the motif-scaffolding problem." arXiv preprint arXiv:2206.04119 (2022).
> > * Gao, Yujuan, et al. "Real-value and confidence prediction of protein backbone dihedral angles through a hybrid method of clustering and deep learning." arXiv preprint arXiv:1712.07244(2017).
> > * AlQuraishi, Mohammed. "End-to-end differentiable learning of protein structure." Cell systems 8.4 (2019): 292-301.
> > * Chowdhury, Ratul, et al. "Single-sequence protein structure prediction using a language model and deep learning." Nature Biotechnology (2022): 1-7.

---

### Official Review · Reviewer_t1c7 · 2022-10-25

**Confidence:** 4
**Correctness:** 1
**Technical Novelty And Significance:** 3
**Empirical Novelty And Significance:** 2
**Recommendation:** 5

**Clarity, Quality, Novelty And Reproducibility:**

### Clarity
The paper is clearly written and structured. Mathematical notation follows standards in the field. Figures are well designed.

### Novelty

The paper introduces a novel representation of proteins in terms of torsional angles and a diffusion process on it. The paper introduces a new problem and task: to generate new foldable protein structures.  As such, these novelties clearly lie within the application area which limits the relevance and significance for the broader ML field.

Q: Can the authors include more references to representations of protein structures? There might be ones that suggest a similar representation.

### Quality


A) Lack of baselines and compared methods. The authors only present a single method and architecture that performs the generative modeling on the proposed representation. However, there are baseline methods that should be compared, e.g. just training a transformer or LSTM in autoregressive (AR) mode to predict the next angles (in other words: using AR models instead of diffusion models). A naive baseline (“BM”) should be the following: sampling one protein structure from the training set and putting minimal amount of noise on the angles. Or some simple baseline that just used empirical distributions of the angles and samples from it. Furthermore, the authors mention several other methods that generate foldable protein structures (Section 2.1): those should definitely be compared against the proposed method.

B) The proposed metric is insufficient because it can easily be tricked with a simple method. The baseline method BM, suggested above, should be good at all proposed metrics: Figure 2-4, and also with respect to developability and scTM score. Thus,  the presented metrics can only be used for a sanity check if the model has learned something useful. A similar problem with evaluation occurs in generating images and this is a fundamental problem of evaluating generative models in continuous spaces or how to assess the amount of novelty. In computer vision, there is the Frechet Inception Distance (FID), with variants for audio and video, and for small molecules the Frechet ChemNet Distance (FCD) that is used to assess unconditional generation. I believe that there is no way around a similar approach to come up with a metric for this task.
The authors should improve the metrics and demonstrate that naive methods cannot reach high performance values on these metrics.

C) Lack of repetitions, confidence intervals, and error bars. It appears that all metrics, figures, and tables show the result of a single run of the method. Therefore, these might arise just by chance and another run could produce a completely different picture. The authors should perform re-runs for their proposed method and all baselines and compared methods.

### Reproducibility
The authors have uploaded code as supplementary data, which should make the work well reproducible

### Minor Comments
- Concerning the statement of designability of the generated structures, it is mentioned that a similar approach as Trippe et al. (2022) was used for evaluation, but Alphafold (Jumper et al., 2021) instead Omegafold (Wu et al., 2022) was used.
- For benchmarking purposes it is preferable to use the same scheme as the method to compare with.
- For Figure 3 it states that the lines are artefacts of null values, there is no discussion why they look shifted for the generated structures, regarding angle phi.
- Comparison between Omegafold (Wu et al., 2022) and Alphafold (Jumper et al., 2021), to have a better benchmark to former works.


### References
[1] Heusel, M., Ramsauer, H., Unterthiner, T., Nessler, B., Klambauer, G., & Hochreiter, S. (2017). Gans trained by a two time-scale update rule converge to a nash equilibrium. arXiv preprint arXiv:1706.08500, 12(1).

[2] Preuer, K., Renz, P., Unterthiner, T., Hochreiter, S., & Klambauer, G. (2018). Fréchet ChemNet distance: a metric for generative models for molecules in drug discovery. Journal of chemical information and modeling, 58(9), 1736-1741.

[3] Ashish Vaswani, Noam Shazeer, Niki Parmar, Jakob Uszkoreit, Llion Jones, Aidan N Gomez, Łukasz Kaiser, and Illia Polosukhin. Attention is all you need. Advances in Neural Information Processing Systems, 30, 2017.

[4] Ruidong Wu, Fan Ding, Rui Wang, Rui Shen, Xiwen Zhang, Shitong Luo, Chenpeng Su, Zuofan Wu, Qi Xie, Bonnie Berger, Jianzhu Ma, and Jian Peng. High-resolution de novo structure prediction from primary sequence. bioRxiv, 2022. doi: 10.1101/2022.07.21.500999.

[5] Brian L Trippe, Jason Yim, Doug Tischer, Tamara Broderick, David Baker, Regina Barzilay, and Tommi Jaakkola. Diffusion probabilistic modeling of protein backbones in 3D for the motifscaffolding problem. arXiv preprint arXiv:2206.04119, 2022.

[6] John Jumper, Richard Evans, Alexander Pritzel, Tim Green, Michael Figurnov, Olaf Ronneberger, Kathryn Tunyasuvunakool, Russ Bates, Augustin Zˇ´ıdek, Anna Potapenko, et al. Highly accurate protein structure prediction with AlphaFold. Nature, 596(7873):583–589, 2021.


**Strength And Weaknesses:**

[+] The representation of the protein backbone as a sequence of angles should be a usefule representation for protein structures
[+] Introduces a new generative learning problem
[+] Using diffusion models to generate foldable proteins is a reasonable approach
[-] Complete absence of informative baselines and other compared methods
[-] Metrics are performed without repetitions and error bars
[-] The metrics can easily be tricked and are insufficient to show that this method works
[-] Low designability of backbones containing beta sheets.
[-] Although there is novelty concerning the framing of the protein backbone as angles, no truly new methods regarding model architecture were established.


**Summary Of The Paper:**

This work introduces a diffusion based approach for unconditional generation of realistic protein structures. It is claimed that the generated structures are similar to naturally-occuring proteins in complexity and structural patterns.

**Summary Of The Review:**

The main novelty of this work lies in the protein structure representation, and the introduction of this generative modeling task. Both novelties are located in an application field. The significance of the work is limited because of the lacking experimental part, with only a single task approached, absence of baselines and naive methods, and insufficient metrics. These same things can also be listed as technical problems, some of which are easier to overcome, e.g. adding baselines and other methods, while others are tricky, e.g. to come up with a good metric.

---

> ### Author Response · Authors · 2022-11-15
> **Response to Reviewer t1c7 (part 1/2)**
>
> Thank you for the insightful and thorough comments. Please see our general comments above, and below, more specific comments to the enumerated concerns.
> ### A. Baselines and comparisons
> We agree having good baseline evaluations is critical to showing the value of generative models. The naive baseline described in point (A), sampling from the empirical distribution of each angle, is included in Appendix A.3 and Figure S3 in our original submission (Appendix C.3 and Figure S7 in our updated manuscript). Briefly, randomly sampling from the empirical distribution of each angle results in no designable proteins. In new results, we have shown that the natural secondary structure co-occurrences illustrated in Figure 4 are also not recapitulated by this random baseline (updated text, new Figure S8). We hope that these results illustrate that our analyses and metrics are not easily undermined by a simple, naive approach, and support the significance of our proposed method and its performance.
>
> Regarding the other mentioned naive baseline – taking an existing protein and noising the angles – this approach does not seem to approximate the structure generation process. For example, taking an alpha helix and noising its angles would simply destroy the alpha helix without arriving at any meaningful alternative structure. We have, however, performed an additional, similar evaluation by taking experimentally determined structures and passing them through the scTM pipeline; we observe that only 80% of natural structures are deemed designable (updated Figure S7).
>
> We appreciate the reviewer’s suggestion to use an autoregressive (AR) model as an additional baseline. We have implemented an AR transformer-based model to predict the next set of angles given all previous angles, which we unroll to generate new proteins using different sets of initial “prompt” angles (see updated text, Appendix C.2). This AR model generates alpha helices exclusively – no beta sheets are detected by P-SEA (updated text, new Figures S5, S6). In other words, the AR collapses to a failure mode where it only generates alpha helices of varying length (qualitatively shown in Figure S5; quantitative distributions shown in Figure S6a), severely compromising the novelty and biological relevance of generated structures. While this model has a higher scTM score (0.29 across 780 generated structures, compared to 0.14 for FoldingDiff’s generation), this is simply because the model is generating highly simplified, repetitive proteins that are of little to no value for actual structure generation. This underscores the importance of our holistic evaluation strategy with multiple metrics, and not just the scTM score (or any other metric) in isolation.
>
> In regards to comparison to methods referenced in our Section 2.1, as we point out in our general response above: no prior diffusion models for protein structure generation make code, model weights, or generated structures available; of non-diffusion structure generation works (as referenced in Section 2.1) only RamaNet, which exclusively generates alpha helices and takes 24hrs for a single structure, has available code. Thus, unfortunately, comparison against these prior works is untenable due to a lack of public code/models and the time and reproducibility constraints for complete re-implementation.
>
> To the reviewer’s question about references to representations of protein structures, we have highlighted several works that use angular representations in generating protein structures: RamaNet (Sabban et al.) and ProteinSGM (Lee et al.). As suggested, we have updated our text to reference additional non-generative works that use an angular representation to predict protein structure from protein sequence (Gao et al., AlQuraishi, Chowdhury et al.).
>
> ### B. Metrics
> Please see our general comment regarding the various metrics used and our ability to compare to prior works. We present an extensive suite of validations – including both those shown by recently proposed protein structure generation works to facilitate comparisons (i.e., scTM), and additional numerical experiments (e.g., secondary structure co-occurrence analyses, Figure 4, new Figure S11) – the extent and depth of which are recognized by both Reviewer Rmta and Reviewer 84e9. As the AR baseline shows, any individual metric may be fragile and subject to trivial solutions, but no trivial solution appears to pass an entire suite of holistic benchmarks. This motivates our usage of a series of validations repeated across multiple random seeds – all of which consistently point to our method producing reasonable results (new Table S1, new Figure S11).
>
> The development of metrics for broad use in this subfield is critically important but not the scope of our work. Furthermore, given prior works’ lack of public code, it is unclear how much information new metrics would provide, given the inability, at present, to directly compare across methods.

---

> > ### Author Response · Authors · 2022-11-15
> > **Response to Reviewer t1c7 (part 2/2)**
> >
> > ### C. Replicate experiments
> > To address concerns surrounding reproducibility and lack of confidence intervals, we have run an additional 5 generations from our model with different random seeds, and evaluated scTM designability and secondary structure co-occurrences for each. These results are shown in the new Table S1 (reproduced below for ease of reference) and new Figure S11. Regardless of random seed, our model’s designability ratios show consistent performance gains compared to prior works; these additional seeds even frequently exceed the performance we report in our primary text (new Table S1). Each random seed also consistently results in structures that contain realistic mixtures of alpha coils and beta sheets (new Figure S11).
> >
> > | Seed | Designable | Designable, short (n=210) | Designable, long (n=570) |
> > | --- | --- | --- | --- |
> > | 1 | 173 | 72 | 101 |
> > | 2 | 154 | 75 | 79 |
> > | 3 | 185 | 90 | 95 |
> > | 4 | 182 | 86 | 96 |
> > | 5 | 187 | 76 | 111 |
> > | 7344 (main text) | 177 | 80 | 97 |
> >
> > ### Novelty
> > While our work is focused on generating novel protein structures, we believe it can be understood much more generally in several ways.
> >
> > We envision that our formulation strategy could be extended as a general approach for modeling and generating paths in 3D space. This could be applied to problems in 3D pose estimation, path generation for robotics and control, or motion refinement, among other applications. Our formulation also represents a more general approach to build representations that are cognizant of the physical or geometric constraints of the corresponding problem, or reduce the solution space to more closely match the space of plausible solutions. Furthermore, one of the key attributes of our formulation is that it learns a representation of each token relative to the state of the prior token. This principle could be applied to other problems, both within and beyond the natural sciences, to potentially enable new advances.
> >
> > Finally, as this paper not only introduces a representation learning strategy but also specifically addresses ICLR’s call for papers in domains including biology (“applications in audio, speech, robotics, neuroscience, biology, or another field”), we believe that our paper presents a valuable contribution for this call and the ICLR community.
> >
> > ### Null value shift in Figure 3
> > As discussed in our manuscript, we subtract the angle-wise mean from each angle during training, and add the mean after generation is fully finished. As an additional implementation detail, nan values are filled with 0 during training after this zero centering has already been done. Therefore, these zero values are shifted to be non-zero after generation, leading to the shift observed in Figure 3b. Figure 3a illustrates the raw experimental angles without any zero centering, and with their nan values directly replaced with 0’s — hence the visual difference.
> >
> > ### Usage of AlphaFold
> > We thank the reviewer for pointing out that our scTM evaluation pipeline is not fully in line with that used by Trippe et al., and thus were previously difficult to compare. We have added new analyses that use ProteinMPNN in place of ESM-IF1 for all analyses in our work (see general comment), and have additionally run AF2 on top of these generated sequences. This now fully matches the scTM procedure used by Trippe et al. and produces a designability of 163/780 – significantly higher than the value of 92/780 reported by this prior work. We hope that these new results aid in evaluating our work relative to past contributions. Unfortunately, due to AF2’s long runtimes, we were not able to analyze all replicates using AF2 as well; however, given AF2’s similarity to OmegaFold in this generated set (163 designable for AF2, 177 using OmegaFold), the existing OmegaFold results should be a good approximation.
> >
> > ### References
> > * Sabban, Sari, and Mikhail Markovsky. "RamaNet: Computational de novo helical protein backbone design using a long short-term memory generative neural network." bioRxiv (2020): 671552.
> > * Lee, Jin Sub, and Philip M. Kim. "ProteinSGM: Score-based generative modeling for de novo protein design." bioRxiv (2022).
> > * Gao, Yujuan, et al. "Real-value and confidence prediction of protein backbone dihedral angles through a hybrid method of clustering and deep learning." arXiv preprint arXiv:1712.07244(2017).
> > * AlQuraishi, Mohammed. "End-to-end differentiable learning of protein structure." Cell systems 8.4 (2019): 292-301.
> > * Chowdhury, Ratul, et al. "Single-sequence protein structure prediction using a language model and deep learning." Nature Biotechnology (2022): 1-7.

---

### Official Review · Reviewer_Rmta · 2022-10-26

**Confidence:** 3
**Correctness:** 4
**Technical Novelty And Significance:** 3
**Empirical Novelty And Significance:** 2
**Recommendation:** 6

**Clarity, Quality, Novelty And Reproducibility:**

Overall, this paper is clear and helpful work for computationally generating novel yet physically foldable protein structures. To my best knowledge, describing protein backbone structure as a series of consecutive angles instead of Cartesian coordinates is a new and promising idea.
Otherwise, authors release the first open-source codebase and trained models for protein structure diffusion which is beneficial for comparison and exchange of other studies.


**Strength And Weaknesses:**

1. This paper proposes a simplified framing of protein backbones . Unlike viewing a protein backbone of amino acids as a cloud of 3D coordinates, authors view it as a sequence of six internal, consecutive angles. The independence of reference frame of each residue leads to no need to use an equivariant neural network. No matter how the protein is rotated or shifted, the angle of the next residue given the current residue never changes.
2. As mentioned before, no requirements of the use of equivariant networks leads to the possibility of using simple transformer as the backbone architecture, thus the model directly generates structures without relying on additional methods for refinement. Generated backbones exhibit greater designability compared to prior works that use equivariance assumptions.
3. It presents a suite of validations to quantitatively demonstrate that unconditional sampling from proposed model directly generates realistic protein backbones – from recapitulating the natural distribution of protein inter-residue angles, to producing overall structures with appropriate arrangements of multiple structural building block motifs.
4. Abundant numerical experiments are supplied to demonstrate that generated backbones contain reasonable structural motifs and they are designable.

**Questions:**
1. The effectiveness of the proposed model has been shown via solid experiments. The potential advantages can be known from an explanation of the paper, but there seems to be less compared with other models in terms of performance and prediction results, could you please provide more valid evidence?
2. Although formulating a protein as a series of angles enables to use simpler models without equivariance mechanisms, this framing allows accumulated errors to significantly alter the overall structure of a generated structure. Could you further analyze the impact of this cumulative error compared to other noise errors based on Cartesian coordinate systems on the prediction results? Does additional refinement similar to the original approach help improve reliability?
3. The generated structures are still simple compared to natural proteins which typically have several hundred residues and it is of static structures. Could you please further explain the possibility and operability of the proposed model in terms of further extensions?



**Summary Of The Paper:**

This paper presents a new diffusion-based generative model that designs protein backbone structures via a procedure that mirrors the native folding process. By considering each residue to be its own reference frame, it describes protein backbone structure as a series of consecutive angles capturing the relative orientation of the constituent amino acid residues instead of Cartesian atom coordinates. A vanilla transformer is used to build a diffusion model to generate new protein structures by denoising from a random, unfolded state towards a stable folded structure. Some experiments indicate resulting model generates lifelike protein, better-respecting protein chirality.

**Summary Of The Review:**

This paper presents a novel parameterization of protein backbone structures that allows for simplified generative modeling. It trains a denoising diffusion probabilistic model with a simple transformer back-bone and demonstrates resulting model unconditionally generates real-life protein structures.

---

> ### Author Response · Authors · 2022-11-15
> **Response to Reviewer Rmta (part 1/2)**
>
> Thank you for the insightful and thorough comments. Please see our general comments above, and below, more specific comments to the enumerated concerns.
>
> ### 1. Additional evidence contextualizing model performance
> As discussed in the general response, a thorough comparison against prior protein diffusion models is not possible due to a lack of public code/models. Instead, we have focused our efforts on improved baseline models. We have added a new autoregressive (AR) baseline that predicts the next set of angles in a structure, given all prior angles. We use several different sets of “seed” angles to generate proteins from this autoregressive model and evaluate generated structures.
>
> We find that this AR baseline collapses to a failure mode of generating alpha helices on endless repeat, regardless of initial “prompt” angles. Please see the new Appendix C.2, Figures S5, S6 in our updated manuscript for additional details. In addition to our previous experiments randomly sampling from the empirical distribution of angles (Appendix C.3), the AR baseline further demonstrates that the task of generating complex proteins with realistic arrangements of secondary structure is highly challenging and cannot be trivially solved using simple approaches. Critically, our method FoldingDiff is not subject to this modal collapse.
>
> We have also run additional generations using different random seeds to validate that our originally presented generations are not the result of a lucky run; these results are in the newly added Table S1 and reproduced below. Across all seeds, the number of designable structures invariably exceeds those reported in prior work, and even outperforms the results discussed in our primary text. These generation replicates all exhibit realistic mixtures of secondary structure elements (new Figure S11).
>
> | Seed | Designable | Designable, short (n=210) | Designable, long (n=570) |
> | --- | --- | --- | --- |
> | 1 | 173 | 72 | 101 |
> | 2 | 154 | 75 | 79 |
> | 3 | 185 | 90 | 95 |
> | 4 | 182 | 86 | 96 |
> | 5 | 187 | 76 | 111 |
> | 7344 (main text) | 177 | 80 | 97 |
>
> We hope these new results provide additional evidence that our method provides robust improvements compared to baseline approaches and performs consistently across independent trials.

---

> > ### Author Response · Authors · 2022-11-15
> > **Response to Reviewer Rmta (part 2/2)**
> >
> > ### 2. Accumulation of error under angular formulation
> > The reviewer is astute in pointing out that our angular formulation potentially allows for errors to propagate through the entire structure. We have included a new series of analyses to better understand this dependency between length and errors (new Appendix A.2, new Figures S2 and S3). First, we analyze the error arising solely from the use of our angular representation. We take natural proteins in their original Cartesian formulation, convert them to angular representation and back to coordinates, and use TMscore to compare the original and reconstructed coordinates. We find that while longer structures tend to have a lower reconstruction TMscore, this TMscore stays around 0.9 even at our maximum size of 128 residues (new Figure S2). Thus, the angular formulation itself does not introduce significant error.
> >
> > Second, we evaluate our model’s ability to learn long range dependencies. We perform an analysis of test set structures, where we add 750 timesteps of noise (out of 1000 total), denoise these mostly-noised examples using our FoldingDiff model, and compare the resulting reconstructed structures to the structures specified by the true value of the angles describing that structure. We find that there is no significant correlation between length and reconstruction error in this experiment, indicating that our model is able to predict the angles themselves accurately regardless of length (new Figure S3).
> >
> > Regarding comparisons to approaches using Cartesian coordinate systems, unfortunately, prior works that apply protein structure diffusion to 3D coordinates do not have code available, and there is a prohibitively large amount of work required for us to implement and optimize such a model ourselves. Direct comparisons to 3D-coordinate diffusion are thus unrealistic at this time.
> >
> > Overall these two new analyses show that our representation itself is minimally lossy at the tested structure lengths and that our model performs well across these lengths.
> >
> > ### 3. Static and short generations compared to natural proteins
> > The reviewer is correct in pointing out that our proteins are relatively short compared to real structures and that our generations are of static structures, while natural proteins are highly dynamic. This is a universal challenge in protein structure generation (motivating a great body of work in molecular dynamics). To generate dynamic structures, one might imagine taking a generated structure, adding a small amount of noise, and repeatedly running the final diffusion steps to approximate a distribution of possible conformations for a given structure. The feasibility and utility of performing such dynamic sampling of structures is left for future work.
> >
> > Overall, we believe that our work and open-source code forms a foundation to build upon with possible extensions in structure lengths, dynamic structures, magnitude of training data, and even the integration of other modalities such as generating amino acid residues corresponding to a structure.

---

### Author Response · Authors · 2022-11-08
**Response to shared concerns among reviewers**

We deeply appreciate the reviewers’ thoughtful and thorough reviews on our work, and for the opportunity to strengthen our manuscript. The reviewers acknowledge the value of our work in developing a diffusion model to generate protein backbones that are realistic and designable, through a novel inter-residue angle formulation. They recognize that our angle-based formulation is distinct from other deep generative models, and appreciate that this framing allows for simplified modeling without equivariance and produces improved generations. The reviewers also generally appreciate that we present a set of well-reasoned, comprehensive analyses of generated structures, and that these analyses are presented with clarity.

Below, we address common concerns shared amongst the reviewers: benchmarks with prior works and baselines, metrics used to evaluate generated structures, and the limitations of our specific angle-based formulation. Over the coming days, we will plan to revise our manuscript with new experimental results that we hope will address reviewer concerns.

# Benchmarks and Comparisons to Prior Works
We agree with the reviewers that additional baselines and comparisons to prior works are valuable to benchmark and contextualize our results.

Reviewer t1c7’s proposal for an autoregressive baseline is an excellent idea. We are currently training an autoregressive transformer for generating a series of angles. We will update both our manuscript and reviewer responses with these results in the coming days.

In addition, as Reviewer t1c7 suggests, randomly sampling from the distribution of natural angles would be informative. This baseline (“naive BM”) was presented and analyzed in Appendix A.3 and Figure S3. Briefly, as shown in Figure S3, random shuffling and sampling of naturally-occurring internal angles results in no designable structures, despite perfectly capturing the overall distribution and pairwise relations between angles.

However, as we point out in our text, no prior diffusion models for protein structure generation make code, pre-trained model-weights, or generated structures publicly available. As pointed out by Reviewer Rmta, our work releases the first open-source codebase and trained models for protein structure diffusion. As such, barring a massive effort to re-implement every single prior work based solely on textual descriptions from the relevant manuscripts (which would invariably produce differing results due to implementation and hyperparameter details), such comprehensive head-to-head comparisons are not feasible. Instead, we have tried to provide comparisons by reproducing figures included in prior works, and quantitatively comparing summary statistics to those previously reported.

Overall, we hope the reviewers can appreciate the limitations involved in benchmarking against prior diffusion models, and that we are doing our best to provide rigorous baselines and comparisons against reported results, given that no public weights or code is available for prior diffusion models. Even among non-diffusion structure generation works we are aware of (e.g., those built on GANs), only RamaNet (Sabban et al.) has available code; however, this method only generates alpha helices (and is thus much simpler) and requires about 24 hours for a single structure, making it unscalable for thorough comparisons. We are also training an autoregressive baseline and have provided the requested comparison to naive BM. As a step towards the goal of greater reproducibility in protein structure design, we are the first deep generative model for protein structure to make all of our code and pre-trained model weights publicly available.

---

> ### Author Response · Authors · 2022-11-08
> **Response to shared concerns among reviewers (cont.)**
>
> # Metrics Used to Evaluate Generated Structures and Replicate Experiments
> We also agree with reviewers that it is difficult to come up with metrics that adequately quantify both the uniqueness and correctness of generated protein backbone structures. This is a near-universal problem for the protein design field at large. As such, while metrics are an evolving area of research, our work follows current practices in protein structure generation. Most recent works use self-consistency TMscores (scTM), and we follow the same approach, showing statistically significant improvements relative to Trippe et al. (Figure 3 in Trippe et al., compared to Figure 5 in our work). Furthermore, we believe we have made an effort to go beyond these standard metrics like scTM, and introduce additional comparisons such as those describing the frequency of secondary structure co-occurrences (see Figure 4).
>
> There is certainly a need to develop better evaluation metrics for protein design. For example, Reviewer t1c7 suggests an analog of the Frechet distance popular in image analysis. This idea has been successful in some molecular domains (e.g., Preuer et al. for small molecules), but an analogous metric for protein structures would require a general protein (backbone) structural embedding method, which to our knowledge has yet to be proposed and widely adopted. Ultimately, we believe that the extensive validation required to formalize an evaluation metric like this would be out of the scope of our present work, though it is certainly something we are considering for future work.
>
> We thank Reviewer t1c7 for the suggestion to perform replicate train-generation experiments and report error bands across multiple trials. We are performing additional replicate experiments (as compute resources allow), including generations with different random seeds, to assess the consistency of our results and our method’s performance. We will also investigate additional scTM calculation pipelines using ProteinMPNN and/or AlphaFold to facilitate more direct comparisons, compute resources permitting.
> # Method Limitations
> The reviewers are correct in pointing out limitations in our method – namely in the accumulation of errors and potential challenges in generalizing to longer sequences that better capture the complexity of real proteins. To make this limitation clearer for readers, we are doing additional experiments to assess reconstruction accuracy across variable lengths, and will update our manuscript with these results in the coming days.
>
> We do note that these limitations are already discussed in our conclusion: e.g., “...this framing allows accumulated errors to significantly alter the overall structure of a generated structure”; “Our generated structures are still of relatively short lengths”; etc. We will update the text to more clearly discuss these limitations. We also acknowledge that our work generates static protein structures, whereas proteins in cells are highly dynamic; this is a universal challenge across all protein generation efforts that have been proposed so far, and one that merits dedicated investigation in future work.
>
> Current research efforts within our team are aimed at addressing these open challenges. However, our work provides the first diffusion-based generative model that generates protein backbone structures using relative internal angles. There are certainly numerous ways in which we could enhance our method, but these methodological improvements are out of scope of the current work, which simply seeks to establish that our angle-based framework is a viable avenue for generation as a first step.
>
> If there are any other standing major concerns from the reviewers, we welcome further discussion and clarifications.
> # References
> * Preuer, Kristina, et al. "Fréchet ChemNet distance: a metric for generative models for molecules in drug discovery." Journal of chemical information and modeling 58.9 (2018): 1736-1741.
> * Trippe, Brian L., et al. "Diffusion probabilistic modeling of protein backbones in 3D for the motif-scaffolding problem." arXiv preprint arXiv:2206.04119 (2022).
> * Sabban, Sari, and Mikhail Markovsky. "RamaNet: Computational de novo helical protein backbone design using a long short-term memory generative neural network." bioRxiv (2020): 671552.

---

> > ### Author Response · Authors · 2022-11-18
> > **Updated regarding ProteinMPNN and AlphaFold2**
> >
> > Based on Reviewer 84e9 and t1c7’s suggestion, we have revisited the use of ProteinMPNN rather than ESM-IF1 for the inverse folding step in our scTM pipeline. We have found that ProteinMPNN in CA mode (we previously used the full atom model) provides significantly improved designability (i.e., for our primary results, from 111/780 to 177/780 designable structures for ESM-IF1 vs. ProteinMPNN, respectively). This trend is consistent across multiple replicate experiments, as suggested by Reviewer t1c7 (Table S1). In light of these new results, we have updated our text to use ProteinMPNN in place of ESM-IF1 for the inverse folding step in assessing backbone designability. Table S1 and Figures 5, 6, S6b, S7, S12, S13, S14, S15 have been updated accordingly. We have added new text that details this difference between inverse folding methods (Appendix B) as well as a new Table S2 that contains the original ESM-IF1 results for generated structures from multiple random seeds.
> >
> > To address Reviewer t1c7’s concerns regarding our use of OmegaFold instead of AlphaFold, we have now run AlphaFold on all the generated structures presented in the primary text. We have found that the designability is highly similar to using OmegaFold (163/780 for AlphaFold, compared to 177/780 for OmegaFold). More importantly, using ProteinMPNN and AlphaFold allows us to do a like-for-like comparison against scTM scores presented by Trippe et al. Our designability scores are significantly higher than this prior work’s proportion of 92/780, with significant improvements in designing both short structures with up to 70 residues, and in designing long structures up to 128 residues.
> >
> > We have updated the text to discuss these new results. Together, our results comprehensively show that FoldingDiff is capable of generating structures that demonstrably improve on prior works. Our method is extensible and should facilitate improved reproducibility and benchmarking, as it is the first open-source deep generative model for protein structure generation.

---

### Author Response · Authors · 2022-12-06
**Invitation for additional discussion**

We have responded to all reviewer critiques with additional analyses to demonstrate the strength of our method except for developing a new FID-like protein metric, which is out of scope for our current work. Given that some time has elapsed for reviewers to revisit our revisions and responses, we greatly look forward to their responses and comments regarding our updated analyses that encompass new baselines, benchmarking with ProteinMPNN and AlphaFold2, and additional replicates. We look forward to engaging with reviewers in this period.

---

### Decision · Program_Chairs · 2023-01-20

**Decision:**

Reject

**Justification For Why Not Higher Score:**

Despite the diffusion model with angular representation is a novel idea in this manuscript, the significance of the work is limited due to the insufficient experimental studies, with only a single task approached, absence of comparative studies, and insufficient metrics. Such concerns are raised by various reviewers, and the authors adopt some suggestions to add new experimental results in response. In a summary, some parts in experimental studies could be significantly improved, so the current manuscript is not ready for publication. If the authors incorporate all the responses properly in revision, then we won't object a acceptance of this paper.

**Justification For Why Not Lower Score:**

N/A.

**Metareview: Summary, Strengths And Weaknesses:**

This manuscript proposes a denoising diffusion probabilistic model for generating protein backbone structures based on torsion angle representations. To overcome the complication of equivariance when working with 3D cartesian coordinate representation, the approach directly models the torsion angles that describe the orientation of residues relative to their neighbors, a rotation invariant representation. Then a denoising diffusion model is applied to such an angular representation to generate new structures. Experimental results show that the model generates diverse structures that recapitulate the test distribution to a large extent. Despite the diffusion model with angular representation is a novel idea, the significance of the work is limited due to the insufficient experimental studies, with only a single task approached, absence of comparative studies, and insufficient metrics. Such concerns are raised by various reviewers, which shows the current manuscript is not ready for publication yet.